# Ca²⁺ waves and ethylene/JA crosstalk orchestrate wound responses in Arabidopsis roots

Xuemin Ma [1], M Shamim Hasan [2], Muhammad Shahzad Anjam [1], Sakil Mahmud [3,4], Sabarna Bhattacharyya [3], Ute C Vothknecht [3], Badou Mendy [2], Florian M W Grundler[2] & Peter Marhavý [1]✉

## Abstract

**Wounding triggers complex and multi-faceted responses in plants. Among these, calcium (Ca²⁺) waves serve as an immediate and localized response to strong stimuli, such as nematode infection or laser ablation. Here, we investigate the propagation patterns of Ca²⁺ waves induced by laser ablation and observe that glutamate-receptor-like channels (GLR3.3/GLR3.6), the stretch-activated anion channel MSL10, and the mechanosensitive Ca²⁺-permeable channels MCA1/MCA2 influence this process. These channels contribute to ethylene-associated signaling pathways, potentially through the WRKY33-ACS6 regulatory network. Furthermore, our findings show that ACC/ethylene signaling modulates Ca²⁺ wave propagation following laser ablation. Ethylene perception and synthesis at the site of damage regulate the local jasmonate response, which displays tissue-specific patterns upon laser ablation. Overall, our data provide new insights into the molecular and cellular processes underlying plant responses to localized damage, highlighting the roles of specific ion channels and hormone signaling pathways in shaping these responses in Arabidopsis roots.**

**Keywords** Ca²⁺ Wave; Ethylene; Jasmonate; Laser Ablation
**Subject Categories** Membranes & Trafficking; Plant Biology; Signal Transduction

## Introduction

Plants cellular damage can be induced by mechanical stress (e.g., wind, crop harvesting), herbivore feeding, and invading microbes triggers. Wounding triggers plant immunity, which is activated by endogenous molecules released from wounded tissue and may act in the form of damage-associated molecular patterns (DAMPs; e.g., peptide systemin, oligogalacturonides, extracellular ATP) (Savatin et al, 2014; Choi and Klessig, 2016). In plants, wounding triggers both local and systemic responses that result in the activation of various cellular processes, including the triggering of Ca²⁺ transients, reactive oxygen species (ROS), signaling cascades involving mitogen-activated protein kinases (MAPKs), hormones, electrical signals, transcriptional reprogramming, and metabolic changes (Marhavý et al, 2019; Wang et al, 2019; Hou et al, 2019; Vega-Muñoz et al, 2020; Zhou et al, 2020).

The calcium (Ca²⁺) wave, in which cells rapidly increase cytosolic free Ca²⁺ levels upon stimuli that spread from cell to cell, integrates multicellular responses (Leybaert and Sanderson, 2012). The regulation of the propagation of the Ca²⁺ wave is complicated and integrated with multiple signaling pathways. ATP can induce the increase of cytosolic Ca²⁺ levels and maintain Ca²⁺ wave transmission over larger distances (Clark and Roux, 2018; Matthus et al, 2020 Donati et al, 2021). The traveling Ca²⁺ wave can be regulated by the diffusion and bulk flow of amino acid, which can activate the calcium-permeable channel GLUTAMATE RECEPTOR-LIKE 3.3 (Bellandi et al, 2022; Grenzi et al, 2023; Alfieri et al, 2020). Plant GLUTAMATE RECEPTOR-LIKE channels (GLRs) act as non-selective cation channels (NSCCs). During mechanically-induced damage of aerial tissues, GLR3.3 and GLR3.6 participate in the Ca²⁺ wave transmission and wound-induced surface potential changes (Mousavi et al, 2013; Toyota et al, 2018; Nguyen et al, 2018). The stretch-activated anion channel MSL10 also contributes to both electrical and Ca²⁺ signaling upon leaf tissue damage, and it works in a parallel pathway as GLRs (Moe-Lange et al, 2021). ROS is involved in Ca²⁺-induced Ca²⁺-release mechanisms (Evans et al, 2016). Decreased Ca²⁺ wave speed has been shown in the Arabidopsis *rbohD* (*respiratory burst oxidase homolog D*) knockout mutant (Evans et al, 2016).

Calcium (Ca²⁺) is one of the earliest signals activated upon wounding and can initiate a signaling cascade that leads to downstream defense responses (Razzell et al, 2013; Ma et al, 2023). For example, cytosolic Ca²⁺ transients are integral to DAMP peptide, Pep-activated plant immune signaling cascades, where cyclic nucleotide-gated channels (CNGCs) play a role in regulating the expression of key defense-related genes like MPK3 and WRKY33 (Ma et al, 2023). The MPK3/MPK6-WRKY33 is a well-known signaling pathway, with WRKY33 serving as a master regulator of downstream plant defense gene expression in response to both biotic and abiotic stresses (Mao et al, 2011; Datta et al, 2015; Wang et al, 2018; Matsumura et al, 2022; Ma et al, 2023). Moreover, the induction of

[1]Umeå Plant Science Centre, Department of Forest Genetics and Plant Physiology, Swedish University of Agricultural Sciences, Umeå 90183, Sweden. [2]Department of Molecular Phytomedizin, Rheinische Friedrich-Wilhelms-University of Bonn, Bonn, Germany. [3]Plant Cell Biology, Institute of Cellular and Molecular Botany, University of Bonn, Bonn, Germany. [4]Present address: Department of Biochemistry, University of Missouri, Columbia, MO 65211, USA. ✉E-mail: peter.marhavy@slu.se

*WRKY33* expression by AtPep peptides can be attenuated by the $Ca^{2+}$ channel blocker $Gd^{3+}$ (Qi et al, 2010), linking the $Ca^{2+}$ signature directly to potential downstream defense responses.

Phytohormones also play a pivotal role in the early response to local tissue damage (Marhavý et al, 2019; Mousavi et al, 2013). Among the hormones involved in plant wounding responses, ethylene has been shown to exhibit a non-systemic response to single-cell ablation (Marhavý et al, 2019). Ethylene plays a critical role in plant defense against pathogens, as evidenced by studies demonstrating that pretreating tomato plants with ethylene reduces their susceptibility to the fungal pathogen *Botrytis cinerea* (Díaz et al, 2002). Furthermore, ethylene acts as an early signal for plant roots to detect soil compaction (Pandey et al, 2021). In contrast, jasmonic acid (JA) is essential for systemic defense (Wang et al, 2019; Nguyen et al, 2018; Glauser et al, 2008). JA triggers plant immune responses (Zhai et al, 2013; De Torres Zabala et al, 2016; Du et al, 2017) and facilitates regeneration after wounding (Zhang et al, 2019; Zhou et al, 2019). Although the regulation of local and systemic wounding responses involves similar cascade processes, each may serve distinct functions. One crucial aspect is the rapid closure of plasmodesmata, which acts as a barrier to limit wound signals reaching undamaged tissues. This closure is facilitated by the generation of reactive oxygen species (ROS) and $Ca^{2+}$ transients at the local level (Vega-Muñoz et al, 2020). However, long-distance signals can also play a significant role in the wound response. Signals such as $[Ca^{2+}]_{cyt}$ (cytosolic free $Ca^{2+}$) and electrical signals can induce rapid leaflet movement in *Mimosa pudica* (Hagihara et al, 2022). This indicates that distant parts of the plant can respond to a wound stimulus through these long-range signals. Interestingly, after mechanical damage ethylene production and perception are involved in the inhibition of JA responses in local but not in distant leaves (Rojo et al, 1999). This suggests that the plant's defensive response can be modulated differently depending on proximity to the injury site.

In this work, we show the $Ca^{2+}$ wave pattern in *glr3.3glr3.6*, *msl10-1*, and *mca1mca2* mutants upon single-cell laser ablation. We also analyzed the function of GLR3.3/GLR3.6, MSL10, and MCA1/MCA2 in the signaling cascade of ethylene responses based on the well-known WRKY33-ACS6 regulation relation. Furthermore, we focus on the early responses of $Ca^{2+}$, ethylene, and JA, delving into their interactions to shed light on their intricate regulation network in promoting local immunity.

# Results

## Loss of GLR3.3/GLR3.6, MSL10, and MCA1/MCA2 alters $Ca^{2+}$ wave patterns following single-cell laser ablation

The infective juveniles of the cyst nematode (*Heterodera schachtii*) pose a significant threat to roots, progressing by sequentially damaging individual cells in the different root layers (Marhavý et al, 2019; Holbein et al, 2019). To investigate whether the presence of attacking cyst nematodes would induce $Ca^{2+}$ waves, we used an Arabidopsis line expressing the genetically encoded $Ca^{2+}$ indicator (GECI), GCaMP3, a single-wavelength, intensity-based GECI (Tian et al, 2009; Nguyen et al, 2018). We visualized $Ca^{2+}$ wave induction resulting from nematode-mediated wounds (resulting from the stylet-breaking cells) through time-lapse confocal imaging (Appendix Fig. S1A–C). Uninfected roots were used as controls. In the control, fluorescence recordings remained stable at a very low level, maintaining a consistent baseline throughout the observation period (Appendix Fig. S1B). However, following cyst

nematode invasion of the roots, we detected an immediate increase in $Ca^{2+}$ levels. Subsequently, these levels decreased and stabilized at a slightly elevated state for the duration of the observation period (Appendix Fig. S1B,C). Our data demonstrate that nematode-induced damage in roots leads to the activation of $Ca^{2+}$ wave propagation, similar to those induced by a single-cell laser ablation (Marhavý et al, 2019). To support the observations of nematode damage-induced $Ca^{2+}$ waves, we further studied the $Ca^{2+}$ wave regulation in more controlled conditions by using a single-cell laser ablation approach. We observed an immediate $Ca^{2+}$ wave when a single cortex cell was ablated in the root of Arabidopsis wild type (Fig. 1A–D). Although the $Ca^{2+}$ wave could propagate in both directions here, we examined the propagation of $Ca^{2+}$ in the root-to-shoot direction (Fig. 1A), to maintain the consistency in our data. The role of $Ca^{2+}$ in regional wounding and signal propagation was assessed on the same side of the root as the ablation site in wild-type seedlings and three mutant lines: *glr3.3glr3.6*, which lacks the glutamate-receptor-like channels GLR3.3/GLR3.6; *msl10-1*, deficient in the stretch-activated anion channel MSL10; and *mca1mca2*, which lacks the $Ca^{2+}$-permeable mechanosensitive channels MCA1 and MCA2. GLR3.3/GLR3 have been implicated in plant defense against herbivores and hormone signaling (Toyota et al, 2018; Nguyen et al, 2018; Wang et al, 2019). MSL10 is involved in plant signaling pathways activated by biotic and abiotic stressors, as well as in programmed cell death (Veley et al, 2014; Basu and Haswell, 2020; Moe-Lange et al, 2021; Basu et al, 2022). MCA1 plays a role in sensing cell wall damage and mechanical stimuli in the root (Nakagawa et al, 2007; Denness et al, 2011). To investigate the impact of these channels on $Ca^{2+}$ signaling, we monitored $Ca^{2+}$ waves following single-cell ablation in *glr3.3glr3.6GCaMP3* (Nguyen et al, 2018), *msl10GCaMP3*, and *mca1mca2GCaMP3* seedlings (Fig. 1B–G). All three mutant lines exhibited a delayed $Ca^{2+}$ wave propagation compared to the WT (Fig. 1B–G; Movies EV1 and EV2). This suggests that GLR3.3/GLR3.6, MCA1/MCA2, and the stretch-activated anion channel MSL10 contribute to the regulation of $Ca^{2+}$ wave dynamics. A similar trend was observed during cyst nematode invasion, where the $Ca^{2+}$ spike induced by nematodes was significantly reduced in *glr3.3glr3.6* compared to WT (Appendix Fig. S1A–C). This finding aligns with the laser ablation data, reinforcing that GLR3.3/GLR3.6 plays a role in facilitating $Ca^{2+}$ wave propagation. In contrast, during nematode wounding, the $Ca^{2+}$ spike was completely absent around 700 s in the mutant compared to the WT (Appendix Fig. S1B), likely due to our measurements being taken at a location further from the nematode-infected region rather than immediately adjacent to it (Appendix Fig. S1B).

## GLR3.3/GLR3.6, MSL10, and MCA1/MCA2 may modulate ethylene responses through the WRKY33-ACS6 signaling pathway

Next, we wanted to further understand the role of GLR3.3/GLR3.6, MSL10, and MCA1/MCA2 in the laser ablation context with downstream hormone responses. Previous data showed that ethylene is robustly regionally induced by single-cell laser ablation in the cortex region but neither Jasmonate nor salicylic acid (Marhavý et al, 2019). We continued investigating the function of GLR3.3/GLR3.6, MSL10, and MCA1/MCA2 upon laser ablation with ethylene responses. *ACS6* promoter-reporter lines have been verified for ethylene responses

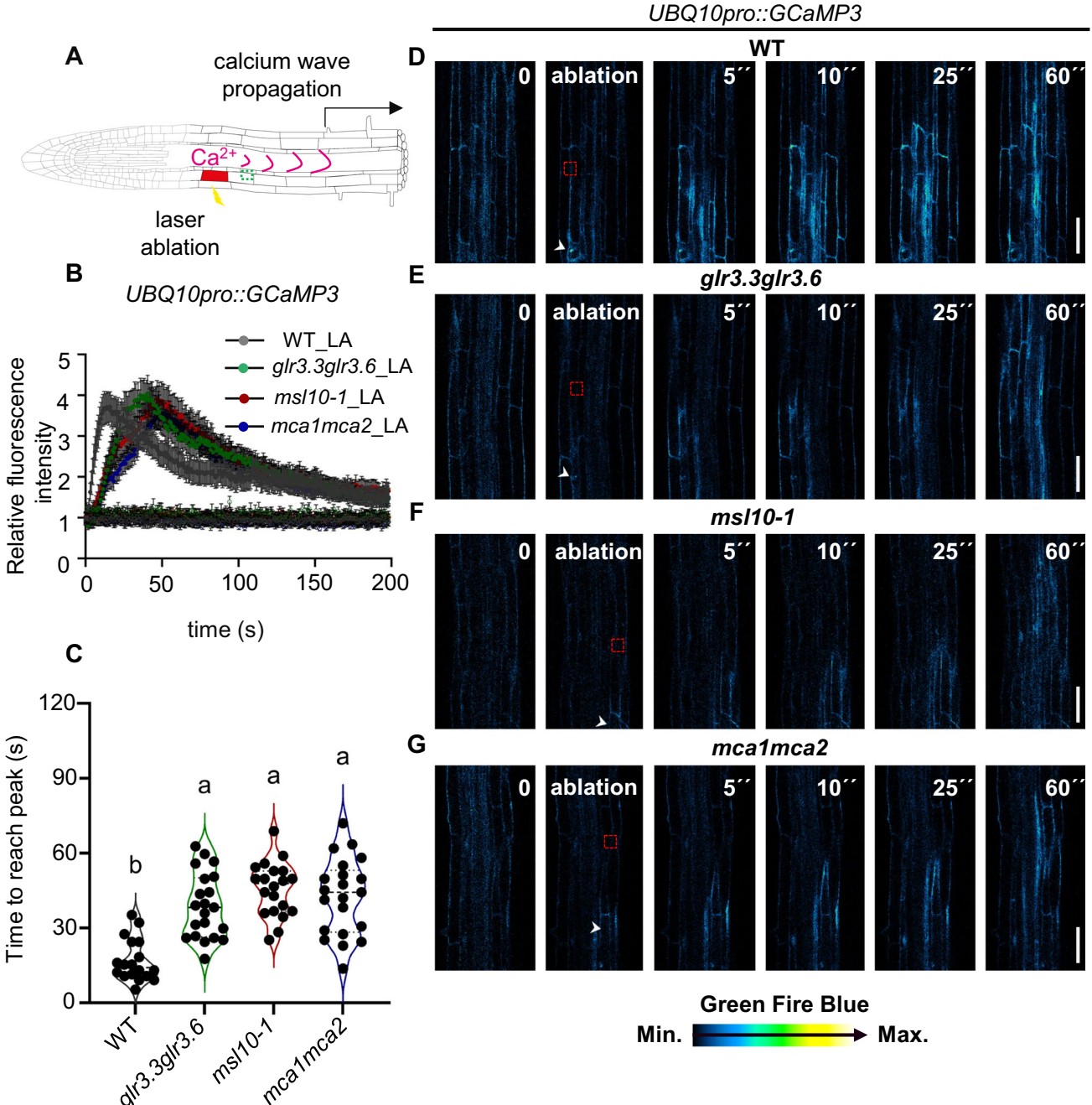

**Figure 1. Loss of GLR3.3/GLR3.6, MSL10, and MCA1/MCA2 alters the Ca²⁺ wave pattern following single-cell laser ablation.**

(A) A schematic diagram depicting single-cell ablation by laser triggering a regional Ca²⁺ wave in Arabidopsis root; the green frame indicates the region of signal quantification. There are calcium waves in directions toward the shoot and root tip region, we select one direction to quantify. (B) Real-time monitoring and quantification of calcium wave propagation after cortex cell ablation (single-cell laser ablation, hereafter LA) using a *UBQ10pro::GCaMP3* fluorescence reporter line in the WT (Col-0) (*n* = 18), *glr3.3glr3.6* (*n* = 21), *msl10-1* (*n* = 20), and *mca1mca2* (*n* = 21) mutants (*n* = three biological pools, each pool including 5–8 seedlings). (C) Calcium speed was quantified as the time it takes for the relative fluorescence intensity of *UBQ10pro::GCaMP3* to reach the maximum for the region (green frame indicated in (A)). (D–G) Representative time-lapse images of WT (D), *glr3.3glr3.6* (E), *msl10-1* (F), and *mca1mca2* (G) expressing *UBQ10pro::GCaMP3* before/after cortex cell ablation are shown. Time points in seconds (s) at the top right corner of each frame. A white arrow indicates the position of the ablated cell, and a red frame indicates the region of signal quantification. Data information: In (B), error bars indicate standard error. In (C), violin plots show the probability density of the data at different values, with wider sections indicating higher density. The dash line represents the median, and dot line represents the quartiles. Different letters (a, b) indicate statistically significant differences between groups (*P* < 0.05, one-way ANOVA followed by Tukey's test) with *P*(b) >0.05 and *P*(a) <0.05. Scale bar: (D–G) 50 μm.

(Poncini et al, 2017; Marhavý et al, 2019). We introduced *ACS6::NLS-3xVenus* reporter line, into *glr3.3glr3.6*, *msl10-1*, and *mca1mca2*. We found that the signal of *ACS6::NLS-3xVenus* after single-cell ablation in the cortex was significantly reduced in all mutants compared to the WT (Fig. 2A,B). In addition, the *ACS6::NLS-3xVenus* signal was significantly diminished following mechanical crushing of large cell populations using tweezers in the *glr3.3glr3.6*, *msl10-1*, and *mca1mca2* mutants compared to the WT(Appendix Fig. S2A–C) and the qRT-PCR further confirmed the transcript level of *ACS6* in *glr3.3glr3.6*, *msl10-1*, and *mca1mca2* upon local wounding is attenuated compared to WT upon local wounding (Appendix Fig. S2D). Thus, our results indicate that ethylene responses upon laser ablation are modulated by calcium channel protein (GLR3.3/GLR3.6, MCA1/MCA2) and anion channel protein (MSL10).

Previously, it has been shown that mitogen-activated protein kinase 6, MPK6 (Liu and Zhang, 2004; Li et al, 2018) function upstream of *ACS6*. We investigated *ACS6* response in *mpk6-2* mutant plants. We introduced *ACS6::NLS-3xVenus* into the *mpk6-2* mutant and single-cell ablation, or crushing a large population of cells with tweezers, did not induce *ACS6* expression in *mpk6-2*, when compared to the WT (Fig. 2B,C; Appendix Fig. S3A,B) in agreement with qRT-PCR results (Appendix Fig. S3C). These results suggest that this well-known MPK6-*ACS6* regulation is still applied in the local wounding response context.

Given that WRKY33 directly activates the expression of *ACS6*, (Li et al, 2012), we aimed to investigate ethylene responses following laser ablation in the *wrky33-1* mutant. In our experiments, single-cell laser ablation induced a strong *ACS6* response in wild-type plants, which was significantly attenuated in the *wrky33-1* mutant, as observed using the *ACS6::NLS-3xVenus* reporter line (Fig. 2B,C). The same trend was observed after crushing a large population of cells with tweezers (Appendix Fig. S3A,B) and the qRT-PCR further confirmed that the transcript level of *ACS6* in *wrky33-1* upon local wounding is compromised compared to WT upon local wounding (Appendix Fig. S3C). Our result suggests WRKY33 participates in transcriptional regulation of the *ACS6* expression upon local wounding.

Next, we examined WRKY33 activation following laser ablation using the *WRKY33::NLS-YFP* reporter line (Ma et al, 2022). Under control conditions, *WRKY33* expression was strongly induced by laser ablation (Fig. 2D,E). Interestingly, this induction was attenuated by GdCl$_3$, an inhibitor of non-selective cation channels (Fig. 2D,E). A similar pattern was observed in root crushing experiments treated with GdCl$_3$ (Appendix Fig. S4A,B). Subsequent qRT-PCR analysis of mRNA from wounded regions revealed that local wounding enhanced *WRKY33* expression, but this induction was significantly reduced following GdCl$_3$ treatment (Appendix Fig. S4C), consistent with findings from Qi et al (2010).

Furthermore, *WRKY33* expression was notably lower in *glr3.3glr3.6*, *msl10-1*, and *mca1mca2* mutants compared to WT upon local wounding (Appendix Fig. S4D). This observation aligns with Ma et al (2023), who reported reduced Pep3-dependent *WRKY33* expression in *cngc2*, *cngc4* (*dnd2*), and *cngc6* mutants. Plant cyclic nucleotide-gated channels (CNGCs) have been implicated in mediating cytosolic Ca$^{2+}$ influx (Ali et al, 2007; Wang et al, 2017; Tian et al, 2019; Duong et al, 2022). Taken together, our data suggest that GLR3.3/GLR3.6, MSL10, and MCA1/MCA2 may play roles in Ca$^{2+}$-dependent processes potentially linked to the WRKY33-ACS6 regulatory network.

Building on these findings and previously shown robust ethylene responses upon cyst nematodes (*Heterodera schachtii*) infections (Marhavý et al, 2019), we were interested to examine the defense-related phenotypes associated with GLR3.3/GLR3.6, MSL10, and MCA1/MCA2. We conducted a nematode infection assay in the *glr3.3glr3.6*, *msl10-1*, and *mca1mca2* mutants. Our findings revealed that both *glr3.3glr3.6* and *msl10-1* mutants exhibited increased susceptibility to nematode infections, suggesting heightened sensitivity to nematode attacks (Appendix Fig. S5). While an increase in nematode numbers was observed in the *mca1mca2* mutant, the difference compared to the wild-type was not statistically significant. These results imply that GLR3.3/GLR3.6 and MSL10 likely play crucial roles in plant defense mechanisms against nematode infections.

## ACC/ethylene pathway gene regulate Ca$^{2+}$ wave propagation upon single-cell ablation

ACC has been reported as the predicted GLR ligand for inducing Ca$^{2+}$ influx in COS-7 mammalian cells expressing the moss *Physcomitrella patens* GLR1 (*PpGLR1*) (Mou et al, 2020). In addition, ACC was shown to enhance Ca$^{2+}$ influx in Arabidopsis ovules, underscoring its significant role in calcium signaling (Mou et al, 2020). These findings motivated us to explore whether ACC/ethylene could affect the Ca$^{2+}$ wave after single-cell laser ablation. Interestingly, in our experiments, the presence of ACC caused a Ca$^{2+}$ wave delay after single cortex cell ablation (we quantify the time it takes to reach the maximum relative fluorescence intensity) (Fig. 3A,B,E,F; Movies EV3 and EV4). To investigate this phenomenon further, we introduced the *UBQ10pro::GCaMP3* into a ACC synthase hextuple mutant (*acs2-1, acs4-1, acs5-2, acs6-1, acs7-1, acs9-1*), as well as the *ETHYLEN OVERPRODUCER 1* (*ETO1*) mutant *eto1-1*. ETO1 is a negative regulator of ACC synthase (ACS). In *eto1-1*, the lack of the negative regulator of ACC synthases resulted in a Ca$^{2+}$ wave propagation similar to that observed by ACC addition upon single-cell ablation, (Fig. 3C, E, F; Movies EV3 and EV4). In the *acs* hextuple mutant, the Ca$^{2+}$ wave showed a similar pattern as in the WT, but with a reduced peak height, (Fig. 3D–F; Movies EV3 and EV4). Using single mutant lines carrying the fluorescence-based Ca$^{2+}$ sensor, R-GECO1 (Keinath et al, 2015) showed that in ethylene receptor mutant *etr1-1*, the speed of the Ca$^{2+}$ wave decreased compared to the WT (in terms of the time required to reach the maximum relative fluorescence intensity) (Fig. 3G; Appendix Fig. S6A,B,E,F). In *ein2-1* Ca$^{2+}$ wave speed delayed compared to WT, and also there is significant enhanced Ca$^{2+}$ influx upon laser ablation (Appendix Fig. S6A–D,G). These results imply the complexity of the existence of a potential interplay between hormones and Ca$^{2+}$, upon local wounding serving to restore equilibrium after wounding (Fig. 3H). The WT; R-GECO1 sensor (Appendix Fig. S6A and WT; GCaMP3 sensor (Fig. 1B) in laser ablation condition, showed similar patterns with regards to the maximum Ca$^{2+}$ peaks. R-GECO1 showed Ca$^{2+}$ dependent increases are higher than GCaMP3 sensor (Zhao et al, 2011).

Ca$^{2+}$ wave propagation upon wounding can be mediated by the release of local amino acids, and glutamate and glycine can trigger GLR3.3-dependent Ca$^{2+}$ wave (Bellandi et al, 2022). In a mouse laser-induced epidermal photodamage study, Ca$^{2+}$ wave propagation highly depends on the release of ATP from the photodamaged cell (Donati et al, 2021). We tested the effects of L-Glu, ATP, and the peptide, PEP1 on the Ca$^{2+}$ wave pattern after laser ablation. Our

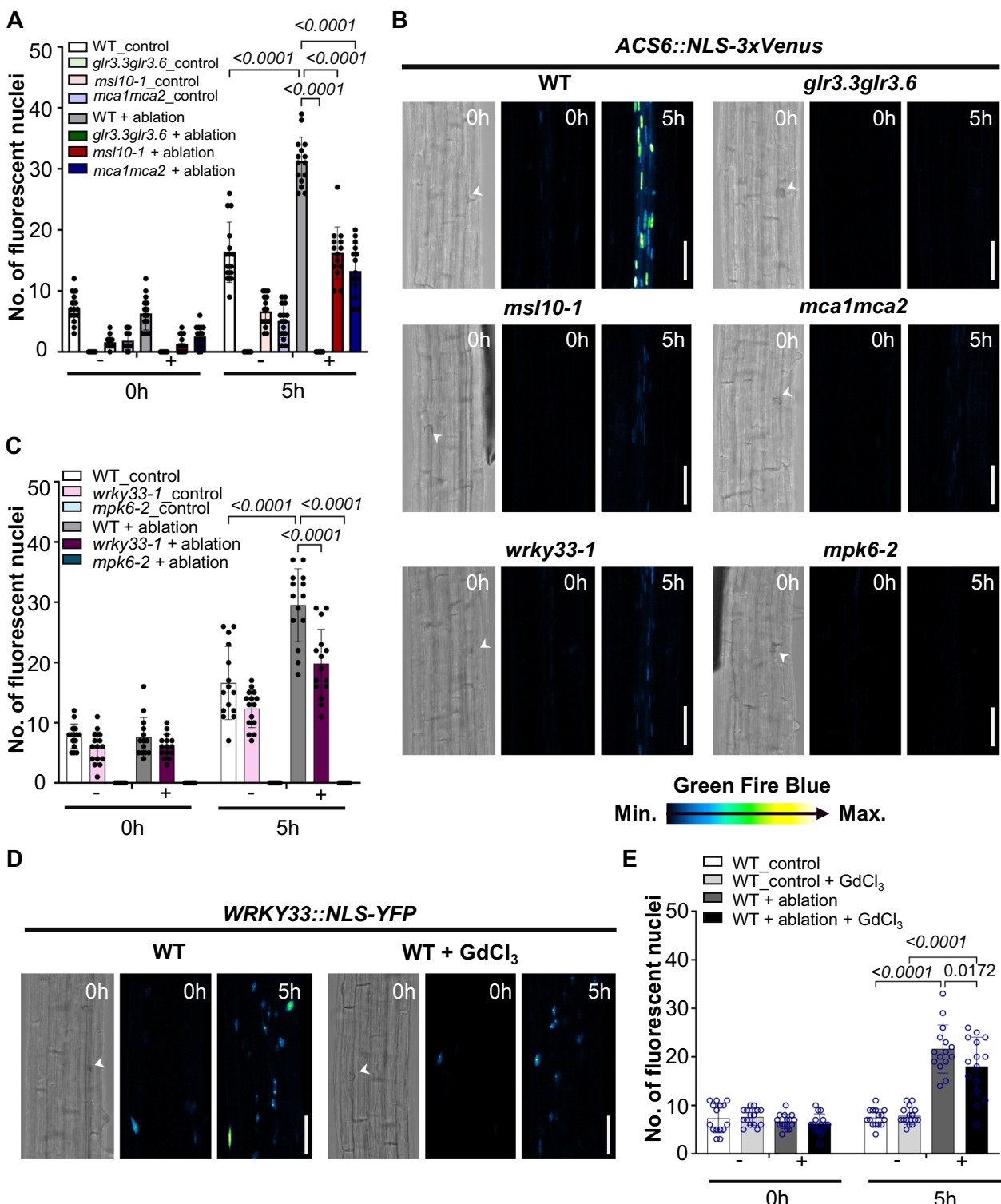

Figure 2. GLR3.3/GLR3.6, MSL10, and MCA1/MCA2 may participate downstream WRKY33-ACS6 signaling pathway upon local wounding.

(A–C) Quantification (A, C) and representative (B) of maximum projection images XYZ of ACS6::NLS-3xVenus in WT (Col-0), glr3.3glr3.6, msl10-1, mca1mca2, wrky33-1, and mpk6-2 at (0 h) or 5 h (h) after laser ablation in the cortex cells. The graph shows a number of cells with a positive nuclear (NLS-3xVenus) signal in individual genotypes. In (A, C), N = three biological pools, each pool includes 4–5 seedlings. (D, E) Quantification (D) and representative (E) of maximum projection images XYZ of WRKY33::YFP-NLS in WT (Col-0) at (0 h) and 5 h after laser ablation (5 h) in the cortex cells, with or without treatment with 50 μM GdCl₃. The graph shows a number of cells with positive nuclear (YFP-NLS) signals. N = three biological pools, each pool includes five seedlings. Data information: In (A, C, E), bars represent mean ± SD. ANOVA Tukey's multiple comparison test was performed with a 95% confidence interval. In (B, D), scale bar: 50 μm. A white arrow indicates the position of the ablated cell. Time points are in hours (h) at the top right corner of each frame.

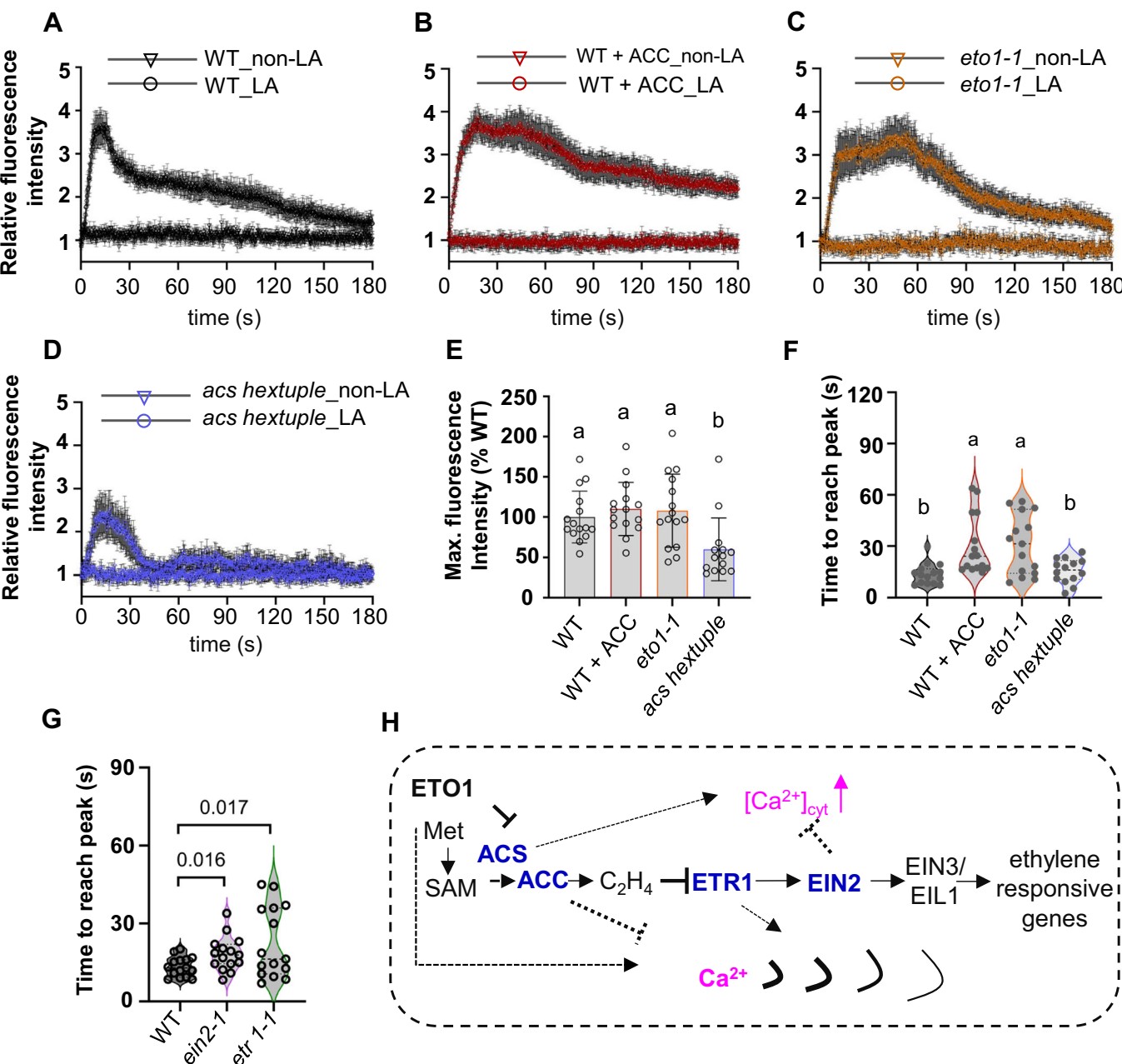

**Figure 3. ACC/Ethylene regulates Ca²⁺ wave propagation upon single-cell laser ablation, model of the ACC/ethylene pathway regulating Ca²⁺ wave propagation.**

(A) Quantification of calcium wave propagation after cortex cell ablation in WT (Col-0) expressing *UBQ10pro::GCaMP3*. (B) Quantification of calcium wave propagation after cortex cell ablation in WT (Col-0) expressing *UBQ10pro::GCaMP3* with the application of 1 μM ACC. (C) Quantification of calcium wave propagation after cortex cell ablation in *eto1-1* expressing *UBQ10pro::GCaMP3*. (D) Quantification of calcium wave propagation after cortex cell ablation in *acs* hextuple mutants expressing *UBQ10pro::GCaMP3* (*acs2-1, acs4-1, acs5-2, acs6-1, acs7-1, acs9-1*). (E) [Ca²⁺]$_{cyt}$ peak indicated by *UBQ10pro::GCaMP3* fluorescence maximum relative intensity after laser ablation. The graph shows average [Ca²⁺]$_{cyt}$ peaks normalized to WT. (F) Calcium speed was quantified as the time it takes for the relative fluorescence intensity of *UBQ10pro::GCaMP3* to reach the maximum for the region (green frame indicated in Fig. 1A). In (A–F), N = three biological pools, each including 4-6 seedlings. (G) Calcium speed was quantified as the time it takes for the relative fluorescence intensity of *R-GECO1* in WT (Col-0), *ein2-1*, and *etr1-1*, to reach the maximum for the region (green frame indicated in Fig. 1A). WT (Col-0); *R-GECO1, ein2-1; R-GECO1, etr1-1; R-GECO1*, N = three biological pools, each pool including five seedlings. (H) The model of the ACC/ethylene pathway regulating Ca²⁺ wave propagation. ACS may positively regulate [Ca²⁺]$_{cyt}$ levels and *EIN2* may negatively regulate it upon single-cell laser ablation. ACC may negatively regulate calcium wave propagation, whereas ETO1 positively regulates it upon single-cell laser ablation. Lines ending with arrows, positive regulation; Lines ending with Ts, negative regulation. Data information: In (A–D), error bars indicate the standard error. In (E), bars represent mean ± SD. In (E, F), different letters (a, b) indicate statistically significant differences between groups (P < 0.05, one-way ANOVA followed by Tukey's test). In (G), Student t test performed, Student's t test; unpaired, two-tailed. In (F, G), violin plots show the probability density of the data at different values, with wider sections indicating higher density. The dash line represents the median, and dot line represents the quartiles.

findings revealed that the initial Ca²⁺ wave caused by laser ablation was delayed in the presence of PEP1 and L-Glu compared to the control (Appendix Fig. S7A,C–F). Interestingly, although ATP treatment had no significant effect on the speed of the Ca²⁺ wave, it did increase the maximum fluorescence intensity compared to the control (Appendix Fig. S7A,B,E,F), demonstrating its critical involvement in the Ca²⁺ wave pattern after laser ablation. P2K1, a purinoceptor, is the receptor for extracellular ATP (eATP). Its mutant, *dorn1*, has been shown to control eATP-dependent cytosolic Ca²⁺ signatures in Arabidopsis roots (Matthus et al, 2020). Interestingly, transcriptomic studies suggest that EIN2-mediated ethylene signaling may be involved in the extracellular ATP response through ROS production (Jewell and Tanaka, 2019). All these findings collectively indicate that ACC/ethylene pathway genes regulate Ca²⁺ wave propagation upon laser ablation and may interact with other pathways, such as ATP signaling.

## JA responds to single-cell ablation in a tissue-specific manner partially depends on ethylene levels

JA is well-described as a wound-associated hormone (Zhai et al, 2013; De Torres Zabala et al, 2016; Du et al, 2017). However, we previously showed that single-cell ablation in roots (especially in the cortex cell) does not induce a robust JA response (Marhavý et al, 2019). To better understand JA responses resulting from single-cell ablation, JA promoter-reporter lines, *AOS::NLS-3xVenus*, and *JAZ10::NLS-3xVenus* were used in our study (Marhavý et al, 2019). As expected, and in agreement with previously published data (Marhavý et al, 2019), none of these markers responded to single cortex cell ablation (Fig. 4A,B; Appendix Fig. S8A).

Previously, it was shown that ethylene production and perception can inhibit the JA response in local leaves while leaving the response in distant leaves unaffected (Rojo et al, 1999). We postulated that the ablation of root cortex cells, leading to ethylene production, similarly might exert an inhibitory effect on JA responses. To test this hypothesis, *AOS::NLS-3xVenus* was introduced into the ethylene mutants *etr1-1, ein2-1, ein3-1, ein3eil1* (Fig. 4C,D; Appendix Fig. S9A). We found a large increase in fluorescence in the *ein3eil1* double mutant indicating that EIN3/EIL1 may negatively regulate *AOS* expression upon single-cell ablation (Fig. 4C,D), which may explain the lack of JA response upon single-cell ablation in the cortex cells (Fig. 4A). Moreover, our data can be explained by data showing that EIN3 can bind directly to the *AOS* promoter and EIN3 perhaps directly regulates *AOS* expression as an upstream regulator (Chang et al, 2013).

To further confirm the potential role of ethylene in suppressing JA-responsive gene expression upon extensive local wounding, we used tweezers to crush root cells in the JA-responsive marker lines. We performed this procedure on both untreated roots and roots treated with the ethylene precursor ACC. We found that *AOS::NLS-3xVenus*, *JAZ10::NLS-3xVenus*, and an additional reporter line, *MYC2::NLS-3xVenus* (Gasperini et al, 2015b) all showed significantly induced responses after such cell damage, but their induction was significantly reduced in the presence of ACC (Appendix Fig. S10A–C). Contrary to ACC treatment, even higher induction of these markers was observed when aminoethoxyvinylglycine (AVG), an ethylene biosynthesis inhibitor, was applied during cell crushing with the exception of the *MYC2* reporter line (Appendix Fig. S10A–C). Moreover, we introduced *JAZ10::NLS-3xVenus* into the ethylene mutants (*etr1-1, ein2-1, ein3eil1, eto1-1*) and performed cell crushing with tweezers

(Appendix Fig. S10D,E). As expected, in *etr1-1*, which is defective in the ethylene receptor, *JAZ10::NLS-3xVenus* expression was significantly induced after local wounding when compared to the wild type. Surprisingly, *JAZ10::NLS-3xVenus* expression was significantly reduced in *ein2-1, ein3eil1,* and *eto1-1* mutants (Appendix Fig. S10D,E). However, this might be explained by a previous observation that 5-day-old seedlings *ein2-1* and *eto1-1* exhibit elevated ethylene levels compared to WT (Woeste et al, 1999). Taken together, JA response to local wounding is partially under the control of ethylene perception and production in roots. Our data aligns with Rojo et al, that ethylene production and perception can inhibit the JA response in local leaves (Rojo et al, 1999).

Interestingly, when we ablated cells at the vascular region within the differentiated part of the root, both *AOS::NLS-3xVenus* and *JAZ10::NLS-3xVenus* were induced (Fig. 4A,B; Appendix Fig. S8B), demonstrating a tissue-specific hormonal response to wounding (Fig. 4F). This result further suggests the existence of a potential inhibitory mechanism employed by nematodes during migration. To delve deeper into this phenomenon, we incubated seedlings of *AOS::NLS-3xVenus* in nematode water (NemaWater) containing a mixture of compounds such as effectors, enzymes, and proteinase (Mendy et al, 2017; Atighi et al, 2020; Goode and Mitchum, 2022). Subsequently, we performed laser ablation within the vasculature and observed a robust reduction of *AOS::NLS-3xVenus* in the presence of NemaWater when compared to the control condition (Fig. 4E; Appendix Fig. S11A,B).

## Wound-induced local JA response depends on the function of GLR3.3/GLR3.6, MSL10, and MCA1/MCA2

Several studies have highlighted the crucial role of GLR3.3/GLR3.6 and MSL10 in JA signaling (Mousavi et al, 2013; Toyota et al, 2018; Wang et al, 2019; Moe-Lange et al, 2021; Bellandi et al, 2022), especially toward systemic wounding response. To understand the regulation of JA responses by GLR3.3/GLR3.6, MSL10, and MCA1/MCA2 upon local wounding, we generated the lines of *AOS::NLS-3xVenus* and *JAZ10::NLS-3xVenus* in *glr3.3glr3.6, msl10-1,* and *mca1mca2*, respectively. Single cortex cell ablation did not induce the expression of *AOS* and *JAZ10* (Fig. 4A,B; Appendix Fig. S8A). We then performed vascular region ablation and did not observe much induction of *AOS::NLS-3xVenus* in *glr3.3glr3.6, msl10-1,* and *mca1mca2* compared to the WT (Fig. 4G,H). As with *AOS::NLS-3xVenus*, the *JAZ10::NLS-3xVenus* signal was reduced in *glr3.3glr3.6, msl10-1,* and *mca1mca2* compared to WT, albeit it is very weak in the WT control (Appendix Fig. S12A). Also, we conducted root crushing experiments on *AOS::NLS-3xVenus/glr3.3glr3.6, AOS::NLS-3xVenus/msl10-1,* and *AOS::NLS-3xVenus/mca1mca2* mutants. The induction of *AOS::NLS-3xVenus* after wounding was compromised in all three mutants (Appendix Fig. S12B). As with *AOS::NLS-3xVenus*, the *JAZ10::NLS-3xVenus* signal was reduced in *glr3.3glr3.6, msl10-1,* and *mca1mca2* (Appendix Fig. S12C). Our results demonstrate that *glr3.3glr3.6, msl10-1,* and *mca1mca2* mutants can attenuate *AOS* and *JAZ10* induction after local wounding when compared to the WT control.

## Discussion

During wounding, regardless of its scale, cytosolic Ca²⁺ responses are immediate, with the highest level closest to the wound site and

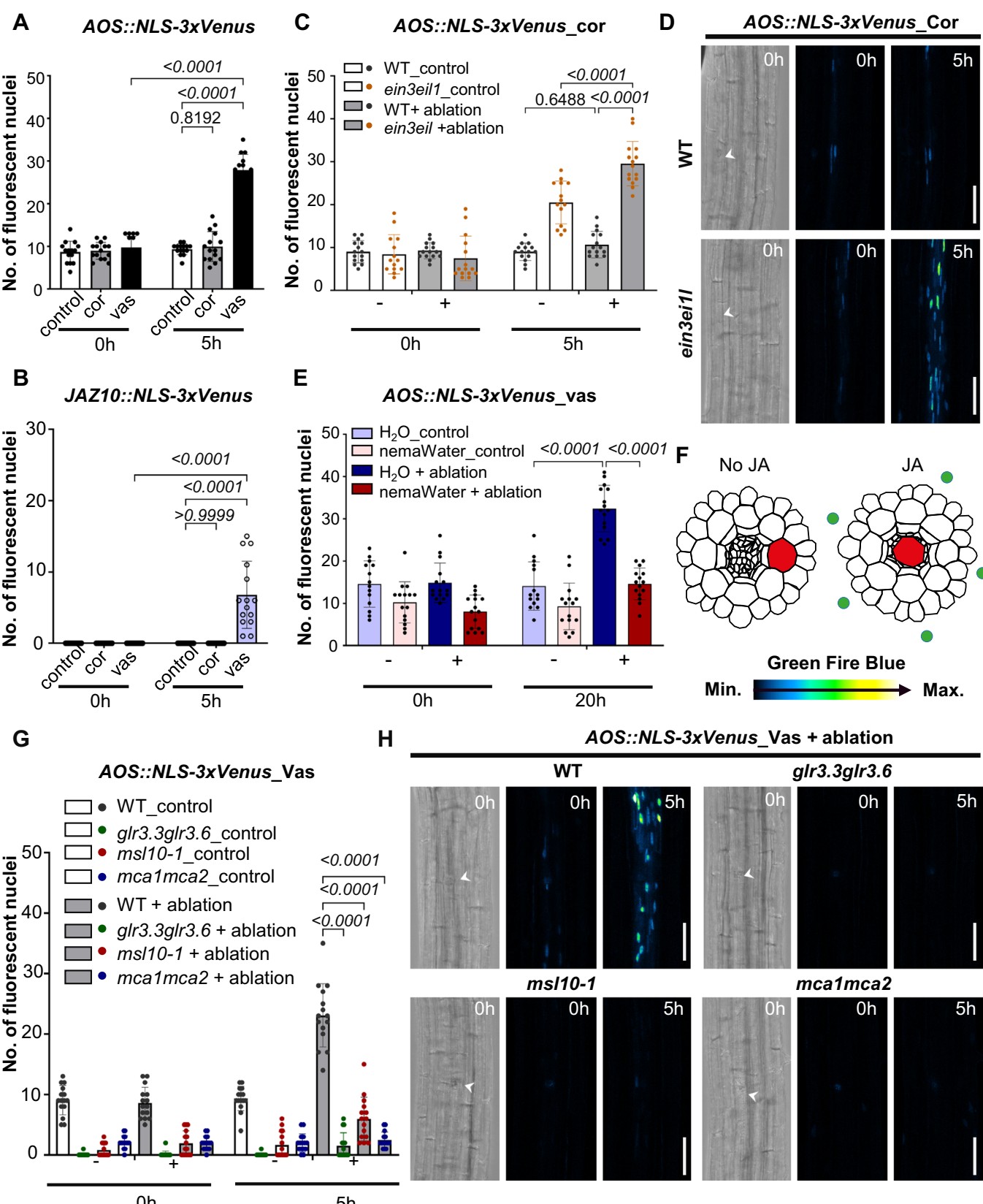

◄ **Figure 4.  JA responds to single-cell ablation in a tissue-specific manner.**

(A, B) Quantification of maximum projection images XYZ of *AOS::NLS-3xVenus* (A), *JAZ10::NLS-3xVenus* (B) the jasmonate-response marker line in WT (Col-0) at  (0 h) and 5 h (h) after laser ablation in the cortex cells (cor), in the vascular region (vas). The graphs show a number of cells with a positive nuclear (NLS-3xVenus) signal. $N$ = three biological pools, each pool includes five seedlings. (C, D) Quantification (C) and representative (D) maximum projection images XYZ of *AOS::NLS-3xVenus* in WT (Col-0) ethylene mutant *ein3eil1* at  (0 h) or 5 h (h) after laser ablation in the cortex cells. The graph shows a number of cells with a positive nuclear (NLS-3xVenus) signal. $N$ = three biological pools, each pool includes 4–5 seedlings. (E) Quantification of maximum projection images XYZ of *AOS::NLS-3xVenus* in WT (Col-0) at (0 h) or 20 h (h) after laser ablation in the vascular region (vas) with pre-treated $H_2O$ for 24 h as mock, or pre-treated nematode water (nemaWater) for 24 h. $N$ = three biological pools, each pool includes 4–5 seedlings. (F) A schematic diagram depicting single-cell ablation by laser in the cortex cell did not induce JA but in the vascular region can induce JA indicated by the green dots. (G, H) Quantification (G) and representative (H) of maximum projection images XYZ of *AOS::NLS-3xVenus* in WT (Col-0), *glr3.3glr3.6*, *msl10-1*, *mca1mca2*, at (0 h) or 5 h (h) after laser ablation in the vascular region. The graphs show a number of cells with a positive nuclear (NLS-3xVenus) signal in all genotypes. Data information: In (A–C, E, G), bars represent mean ± SD. ANOVA Tukey's multiple comparison tests were performed with a 95% confidence interval. In (D, H), time points are in hours (h) at the top right corner of each frame. A white arrow indicates the position of the ablated cell and a scale bar indicates 50 μm.

remaining elevated for a longer duration (Marhavý et al, 2019; Nguyen et al, 2018; Costa et al, 2017; Toyota et al, 2018; Hander et al, 2019). Moreover, in response to herbivore attacks, a rapid induction of $Ca^{2+}$ takes place, contributing significantly to signaling cascades and defense mechanisms (Toyota et al, 2018; Nguyen et al, 2018; Hagihara et al, 2022). Although well-documented above ground, there is a paucity of information regarding wound-induced $Ca^{2+}$ dynamics below ground. This study presents, for the first time, a robust $Ca^{2+}$ wave triggered by nematode-induced cell damage. The $Ca^{2+}$ wave initiated by nematodes results in an immediate increase in $Ca^{2+}$ cytosolic levels near the damage site. To comprehensively examine the dynamics of $Ca^{2+}$ waves, we employed multiphoton confocal microscopy to selectively ablate cells, allowing us to subsequently image and monitor the progression of the fluorescence wave from genetically encoded fluorescent $Ca^{2+}$ sensors across neighboring cells. Through the application of this methodology across various mutants, we identified that *glr3.3glr3.6*, *msl10-1*, and *mca1mca2* mutants exhibited a delayed peak in local $Ca^{2+}$ elevation. These findings align with the hypothesis proposed by Bellandi et al (2022), that the wounding of veins results in the release of amino acids, subsequently activating GLRs, which, in turn, contribute to local calcium signaling. Also, it supports that MSL10 functions in the same pathway with GLRs (Moe-Lange et al, 2021). MSL10 has been shown to involve in a transient calcium influx in response to cell swelling (Basu and Haswell, 2020). MCA1/MCA2 were shown to mediate $Ca^{2+}$ uptake in plants, and they are responsible for the calcium influx in yeast (Nakagawa et al, 2007; Yamanaka et al, 2010). Our data showed that MCA1/MCA2, also participates in $Ca^{2+}$ wave propagation upon laser ablation, and together with the result of GLR3.3/GLR3.6 and MSL10, it indicates that the immediate $Ca^{2+}$ levels elevation in the undamaged cell may depend on multiple players. Despite our findings, when measuring the $Ca^{2+}$ wave on the same side as the ablation, we observed no significant difference among the different backgrounds in terms of the maximum cytosolic $Ca^{2+}$ peak (Fig. 1B). However, this might not be the case if the region of interest (ROI) were shifted to the opposite side of the root. In addition, GCaMPs and R-GECO1 do not exhibit a linear response, meaning that beyond a certain threshold of $Ca^{2+}$ concentration, fluorescence changes plateau, making relatively small differences undetectable. This limitation suggests that maximum $Ca^{2+}$ peaks might still differ. A more precise approach to resolving this issue could be the use of FRET-based ratiometric indicators

(Liese et al, 2024; Lin et al, 2025) which would enhance sensitivity and provide more quantitative insights.

1-aminocyclopropane-1-carboxylic acid synthase 6 (ACS6) serves as a key enzyme in ethylene biosynthesis and is phosphorylated by MPK6, which in turn influences ethylene production (Liu and Zhang, 2004). In addition, MPK6 can regulate *ACS6* at the transcriptional level, a finding consistent with our data showing that *ACS6* expression is downregulated in the *mpk6-2* mutant upon local wounding compared to the WT (Fig. 2B,C; Appendix Fig. S3). WRKY33 is a downstream target of MPK3/MPK6 (Mao et al, 2011; Wang et al, 2018) and has been shown to directly bind to W-box motifs in the *ACS6* promoter through chromatin immunoprecipitation (ChIP) assays (Li et al, 2012). In our study, WRKY33 is the positive upstream regulator of *ACS6* upon local wounding, *ACS6*, upon laser ablation is significantly inhibited in the *wrky33-1* mutant (Fig. 2B,C). *WRKY33* expression can also be induced upon laser ablation (Fig. 2D,E). Moreover, in *glr3.3glr3.6*, *msl10-1*, and *mca1mca2* mutants, *WRKY33* expression is markedly reduced compared to WT upon local wounding in the qRT-PCR result (Appendix Fig. S4D). The direct downstream target of WRKY33, *ACS6* showed attenuated expression in *glr3.3glr3.6*, *msl10-1*, and *mca1mca2* upon laser ablation or local wounding by crushing roots (Fig. 2A,B; Appendix Fig. S2). It indicates that GLR3.3/GLR3.6, MSL10, and MCA1/MCA2 modulate ethylene responses in the laser ablation context, and this may be through WRKY33-*ACS6* regulation relation.

Ethylene plays a pivotal role in regulating plant growth, development (Dubois et al, 2018), and responses to environmental stress (Díaz et al, 2002; Hartman et al, 2019). We now demonstrated that ACC/ethylene changes the $Ca^{2+}$ wave upon single-cell ablation, suggesting a comprehensive interplay between hormones and $Ca^{2+}$ to maintain a balance after wounding (Fig. 3; Appendix Fig. S6). These results align with existing literature, where Mou et al (2020) demonstrated that ACC-induced GLR signaling is evolutionarily conserved in the context of cell-cell communication in land plants. It also aligned with finding that both ethephon, the ethylene-releasing compound, and ACC can activate activated $Ca^{2+}$-permeable cation channels in tobacco suspension cells by using the whole-cell patch-clamp technique. The ethephon-induced $Ca^{2+}$ elevation is compromised when adding $Ca^{2+}$-channel blocker or chelator, supporting ethylene modulates influx of $Ca^{2+}$ of through ethephon-activating $Ca^{2+}$-permeable channels (Zhao et al, 2007). Under salt condition,

ethylene signaling mutant *etr1-1* showed less Ca²⁺ influx than WT by analyzing the steady-state ionic flux kinetics (Lang et al, 2020). Taken together, ACC/ethylene pathway genes may function directly with Ca²⁺-permeable channels to regulate Ca²⁺ homeostasis.

Ca²⁺ wave propagation upon wounding is a complex process influenced by multiple factors. Recent research has shown that the release of amino acids from damaged cells can regulate local Ca²⁺ wave propagation (Bellandi et al, 2022). In this study, we examined the effects of various damage-associated molecules on local Ca²⁺ waves. Under our experimental conditions, treatments with L-Glu and PEP1 delayed the propagation of the Ca²⁺ wave following laser ablation, though the underlying reasons remain unclear. We speculate that our pre-treatments may induce secondary effects that alter the typical Ca²⁺ wave response to laser ablation. Interestingly, ATP treatment enhanced calcium influx upon laser ablation, consistent with findings from a similar "laser ablation" study in mice, which demonstrated the crucial role of ATP release from photodamaged cells in Ca²⁺ wave propagation (Donati et al, 2021). ATP signaling is also interconnected with the ACC/ethylene pathway. In the *ein2-1*, increased expression of the ATP receptor P2K1 (Jewell and Tanaka, 2019) may explain the heightened Ca²⁺ influx observed in *ein2-1* during laser ablation. EATP crosstalk with ethylene response with respect to reduce hypocotyls length (e.g., the eATP effect on hypototyl elongation was abolished in *etr1-1* and *ein3-1eil1-1* loss-of-function mutants (Lang et al, 2020). Furthermore, ATP signaling is closely linked to both ROS and calcium signaling (Tanaka et al, 2010). In the context of laser ablation, it becomes challenging to isolate the effects of individual signaling pathways. ROS production, which occurs upon laser ablation (Marhavý et al, 2019), interacts with ethylene signaling. Ethylene promotes ROS production (Martin et al, 2022). ROS activate the Ca²⁺ channel and aids in the propagation of Ca²⁺ waves (Kwak et al, 2003; Foreman et al, 2003; Evans et al, 2016). It is possible that in ethylene mutants the alteration of ROS levels changed the Ca²⁺ channel activity and displayed varied Ca²⁺ wave pattern upon laser ablation. Taken together, the regulation of Ca²⁺ wave propagation by ACC/ethylene during laser ablation likely results from the interplay of multiple signaling pathways, reflecting the complexity of cellular responses under these conditions.

JA is involved in wound response and plant defense against herbivores and pathogens (Toyota et al, 2018; Grebner et al, 2013; Gasperini et al, 2015a; Kurenda et al, 2019). Notably, while JA has been implicated in controlling the regeneration of the root apical meristem (Zhou et al, 2019), it is important to consider that JA can be transmitted from shoots to roots in response to nematode attack (Wang et al, 2019). Here we provide compelling evidence that the JA responses following laser ablation are intricately tied to tissue-specific induced damage, specifically vascular cell ablation. Furthermore, our findings reveal a novel aspect of the interplay between plant–nematode interactions. During the nematode infection, nematodes may release compounds with a pronounced inhibitory effect on JA responses. Although detailed chemical compounds analysis has not been done yet for nemaWater, we indeed found it has a strong effect on *AOS* response upon vascular region ablation (Fig. 4E). This explains the different results on JA responses during cyst nematode infection, especially in later stages

of parasitism, with some studies suggesting a contribution to susceptibility rather than resistance (Ozalvo et al, 2014). Our data also suggests a potential inhibitory role of ethylene on JA responses. When roots were locally wounded by crushing, the ethylene-overproducing mutant *eto1-1*, and *ein2-1* which both showed enhanced ethylene level at seedling stage (Woeste et al, 1999) showed reduced expression of *AOS* and *JAZ10* compared to the wild-type (Appendix Figs. S9B, S10D, and S10E). In *ein3eil1*, we observed enhanced *AOS* expression when ablation was performed in the cortex cells. This finding might explain why a robust *AOS* response was not detected in the cortex cell ablation. A ChIP-seq study by (Chang et al, 2013) identified *AOS* as one of the target genes of EIN3, supporting the notion that ethylene signaling directly influences JA pathways. Moreover, Rojo et al (1999) demonstrated that locally produced and perceived ethylene negatively regulates JA responses, although this inhibition is alleviated in distal regions, further corroborating our results. In *ein3eil1*, by crushing the roots, *JAZ10* expressed lower than WT (Appendix Fig. S10D). EIN3 interacts with multiple JA pathway genes at protein level, such JAZ1, JAZ3, JAZ9 and MYC2 (Zhu et al, 2011; Song et al, 2014). This causes the JA-ET interaction to be complex. In the scenario of plant defense against the necrotrophic pathogen, *B. cinerea*, it may coordinate both synergistic and antagonistic mechanisms. The derepression of EIN3/EIL1 through interacting with JAZ protein will positively regulate pathogen-related gene expression. While JA-induced JAZ degradation will release MYC2, which also interacted with EIN3, and EIN3/EIL1 attenuate the transcriptional activation function of MYC2 and will lead to repress excessive plant defense responses (Song et al, 2014). JAZ10 is regulated by MYC2 (Van Moerkercke et al, 2019), perhaps its expression upon local wounding is modulated by EIN3/EIL1-MYC2 repression regulation.

Moreover, we confirmed local JA responses upon vasculature ablation, depending on the function of GLR3.3/GLR3.6, MSL10, MCA1/MCA2 (Fig. 4G,H; Appendix Fig. S12A). The promoter activities of GLR3.3/GLR3.6, is more active in vasculature (Nguyen et al, 2018), GLR3.3 is localized to the phloem in leaves while GLR3.6 localization to the contact cells of the xylem parenchyma (Toyota et al, 2018). MSL10 expression overlapped with GLR3.3/GLR3.6, across the vasculature, including both phloem and xylem (Moe-Lange et al, 2021). MCA1/MCA2 promoter also showed more expression in vascular tissues (Yamanaka et al, 2010). This vascular region-specific expression pattern of GLR3.3/GLR3.6, MSL10, MCA1/MCA2 may partially contribute to the JA response to vascular ablation.

In summary, our study elucidated the early events following wounding, specifically the regulation of Ca²⁺ wave propagation and the potential role of the GLR3.3/GLR3.6, MSL10, and MCA1/MCA2 in downstream ethylene signaling. We also uncovered a potential inhibitory effect of ethylene on JA responses during local wounding (Fig. 5), with the JA response exhibiting tissue-specific characteristics and possible hormonal regulation of plant–nematode interactions. Understanding the temporal and spatial regulation of hormones and their crosstalk with upstream calcium signaling remains a challenge, but advancements in research methods will likely provide deeper insights into these complex regulatory networks.

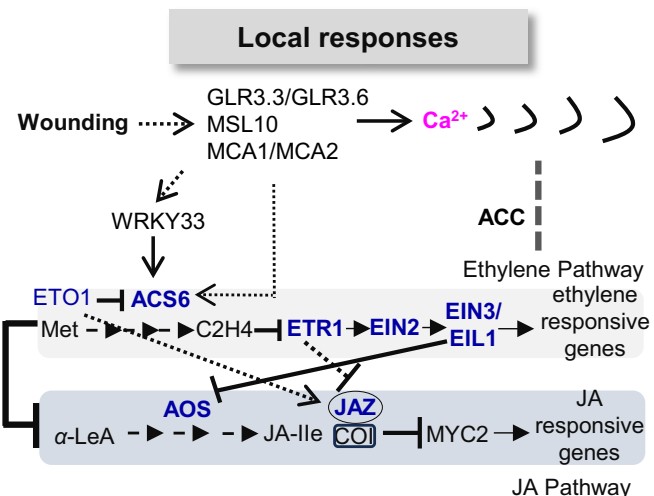

**Local responses**

**Figure 5. A model for the local Ca²⁺ wave, ethylene, and JA responses upon wounding.**

The single-cell ablation in the cortex cell will induce regional Ca²⁺ wave and hormone responses. GLR3.3/GLR3.6, MSL10, and MCA1/MCA2 modulate the extent of Ca²⁺ wave propagation upon laser ablation. GLR3.3/GLR3.6, MSL10, and MCA1/MCA2 may participate in the ethylene response through the well-known WRKY33-*ACS6* regulation relation. Ethylene perception and synthesis at the site of damage seem to influence the local jasmonate response, which exhibits tissue-specific patterns upon laser ablation. ACC and ethylene regulate Ca²⁺ wave propagation upon laser ablation. Lines ending with arrows, positive regulation; lines ending with Ts, negative regulation.

# Methods

### Reagents and tools table

| Reagent/resource | Reference or source | Identifier or catalog number |
|---|---|---|
| **Experimental models** | | |
| *Arabidopsis thaliana*: WT Col-0 | NASC | NCBI:txid3702 |
| Arabidopsis: *UBQ10pro::GCaMP3*; WT | Nguyen et al, 2018 | Transgenic *UBQ10pro::GCaMP3* in Col-0 |
| Arabidopsis: *glr3.3glr3.6* | Nguyen et al, 2018 | N/A |
| Arabidopsis: *glr3.3glr3.6GCaMP3* | Nguyen et al, 2018 | N/A |
| Arabidopsis: *msl10-1* | Moe-Lange et al, 2021 | N/A |
| Arabidopsis: *msl10-1; GCaMP3* | This paper | Transgenic *UBQ10pro::GCaMP3* in *msl10-1* |
| Arabidopsis: *mca1mca2* | Yamanaka et al, 2010 | N/A |
| Arabidopsis: *mca1mca2;GCaMP3* | This paper | Transgenic *UBQ10pro::GCaMP3* in *mca1mca2* |
| Arabidopsis: *acs2-1, acs4-1, acs5-2, acs6-1, acs7-1, acs9-1GCaMP3* | *acs2-1, acs4-1, acs5-2, acs6-1, acs7-1, acs9-1 (acs hextuple)* from (Mou et al, 2020) | Transgenic *UBQ10pro::GCaMP3* in *acs hextuple* |
| Arabidopsis: *eto1-1GCaMP3* | *eto1-1* from (Guzmán and Ecker, 1990) | Transgenic *UBQ10pro::GCaMP3* in *eto1-1* |

| Reagent/resource | Reference or source | Identifier or catalog number |
|---|---|---|
| Arabidopsis: *R-GECO1*; WT (Col-0) | Keinath et al, 2015 | N/A |
| Arabidopsis: *ein2-1;R-GECO1* | *ein2-1* from Marhavý et al, 2019 | Transgenic R-GECO1 in *ein2-1* |
| Arabidopsis: *etr1-1;R-GECO1* | *etr1-1* from Marhavý et al, 2019 | Transgenic R-GECO1 in *etr1-1* |
| Arabidopsis: *AOS::NLS-3xVenus*; WT (Col-0) Arabidopsis: *AOS::NLS-3xVenus; glr3.3glr3.6* | Marhavý et al, 2019 This paper | N/A Transgenic *AOS::NLS-3xVenus* in *glr3.3glr3.6* |
| Arabidopsis: *AOS::NLS-3xVenus; msl10-1* | This paper | Transgenic *AOS::NLS-3xVenus* in *msl10-1* |
| Arabidopsis: *AOS::NLS-3xVenus; mca1mca2* | This paper | Transgenic *AOS::NLS-3xVenus* in *mca1mca2* |
| Arabidopsis: *AOS::NLS-3xVenus; etr1-1* | This paper | Transgenic *AOS::NLS-3xVenus* in *etr1-1* |
| Arabidopsis: *AOS::NLS-3xVenus; ein2-1* | This paper | Transgenic *AOS::NLS-3xVenus* in *ein2-1* |
| Arabidopsis: *AOS::NLS-3xVenus; ein3-1* | This paper | Transgenic *AOS::NLS-3xVenus* in *ein3-1* |
| Arabidopsis: *AOS::NLS-3xVenus; ein3eil1* | This paper | Transgenic *AOS::NLS-3xVenus* in *ein3eil1* |
| Arabidopsis: *JAZ10::NLS-3xVenus*; WT (Col-0) | Marhavý et al, 2019 | N/A |
| Arabidopsis: *JAZ10::NLS-3xVenus; glr3.3glr3.6* | This paper | Transgenic *JAZ10::NLS-3xVenus* in *glr3.3glr3.6* |
| Arabidopsis: *JAZ10::NLS-3xVenus; msl10-1* | This paper | Transgenic *JAZ10::NLS-3xVenus* in *msl10-1* |
| Arabidopsis: *JAZ10::NLS-3xVenus; mca1mca2* | This paper | Transgenic *JAZ10::NLS-3xVenus* in *mca1mca2* |
| Arabidopsis: *JAZ10::NLS-3xVenus;etr1-1* | This paper | Transgenic *JAZ10::NLS-3xVenus* in *etr1-1* |
| Arabidopsis: *JAZ10::NLS-3xVenus;ein2-1* | This paper | Transgenic *JAZ10::NLS-3xVenus* in *ein2-1* |
| Arabidopsis: *JAZ10::NLS-3xVenus;ein3eil1* | This paper | Transgenic *JAZ10::NLS-3xVenus* in *ein3eil1* |
| Arabidopsis: *JAZ10::NLS-3xVenus;eto1-1* | This paper | Transgenic *JAZ10::NLS-3xVenus* in *eto1-1* |
| Arabidopsis: *ACS6::NLS-3xVenus*; WT (Col-0) | Marhavý et al, 2019 | N/A |
| Arabidopsis: *ACS6::NLS-3xVenus; glr3.3glr3.6* | This paper | Transgenic *ACS6::NLS-3xVenus* in *glr3.3glr3.6* |
| Arabidopsis: *ACS6::NLS-3xVenus; msl10-1* | This paper | Transgenic *ACS6::NLS-3xVenus* in *msl10-1* |
| Arabidopsis: *ACS6::NLS-3xVenus; mca1mca2* | This paper | Transgenic *ACS6::NLS-3xVenus* in *mca1mca2* |
| Arabidopsis: ACS6::NLS-3xVenus; *wrky33-1* | *wrky33-1* from NASC, SALK_006603 | Transgenic *ACS6::NLS-3xVenus* in *wrky33-1* |
| Arabidopsis: ACS6::NLS-3xVenus; *mpk6-2* | *mpk6-2* from NASC, SALK_073907 | Transgenic *ACS6::NLS-3xVenus* in *mpk6-2* |
| Arabidopsis: *WRKY33::NLS-YFP*; WT (Col-0) | Ma et al, 2022 | N/A |

| Reagent/resource | Reference or source | Identifier or catalog number |
|---|---|---|
| **Oligonucleotides** | | |
| Primer for qRT-PCR, housekeeping gene UBC1, Forward TCTCTCCGCGATC TTTACCTCAAC Reverse GCGTCGACATCC TCCTTTCTTTCG | This paper | N/A |
| Primer for qRT-PCR, WRKY33 Forward CTTCC ACTTGTTTCA GTCCCTCTC Reverse CTGTGGTTGGAG AAGCTAGAACG | This paper | N/A |
| Primer for qRT-PCR, ACS6 Forward CCAGGGTTTG ATAGAGATTTG Reverse CCGGTCTAA CGTCGTACCAA | This paper | N/A |
| **Chemicals, enzymes, and other reagents** | | |
| GdCl₃ gadolinium chloride | Sigma-Aldrich/Merck | G7532-5G |
| ACC, 1-aminocyclopropanec-arboxylic acid | Sigma-Aldrich | A3903-250mg |
| AVG, A6685 | aminoethoxyvinylglycine | Sigma-Aldrich/Merck |
| PI, propidium iodide | Thermo Fisher | P1304MP-100mg |
| L-glutamate | Sigma-Aldrich | N/A |
| ATP, adenosine triphosphate | Sigma-Aldrich | N/A |
| plant elicitor peptide, AtPEP1 | Zhou et al, 2020 | N/A |
| RNeasy Plant Mini Kit | QIAGEN | 74904 |
| iScript cDNA Synthesis Kit | Bio-Rad Laboratories AB | 1708890 |
| iTaq Univer SYBR Green SMX 200 | Bio-Rad Laboratories AB | 1725120 |
| **Software** | | |
| Fiji (Image J) | https://imagej.net/ij/ | N/A |
| GraphPad Prism 10.4.0 | https://www.graphpad.com/ | N/A |
| Adobe Illustrator | https://www.adobe.com/ | N/A |
| Adobe Premiere Pro | https://www.adobe.com/ | N/A |
| **Other** | | |
| *Cyst* nematode: *Heterodera schachtii* (Bonn population) | MPM, INRES, University of Bonn | N/A |

## Growing Arabidopsis seedlings

The transgenic *Arabidopsis thaliana* lines used in this study were in Col-0 background and included the following: *ACS6::NLS-3xVenus, JAZ10::NLS-3xVenus, AOS::NLS-3xVenus* (Marhavý et al, 2019); *MYC2::NLS-3xVenus* (Gasperini et al, 2015b); *WRKY33::NLS-Venus* (Ma et al, 2022); *R-GECO1* (Keinath et al, 2015); and *UBQ10pro::GCaMP3* (Nguyen et al, 2018). For the selection of positive transgenic plants with *ACS6::NLS-3xVenus, JAZ10::NLS-*

*3xVenus*, and *AOS::NLS-3xVenus*, we used BASTA selection on solid ½ MS media containing 15 mg/L. The mutants used in this study were: *ein2-1; ein3-1; etr1-1, ein3eil1* (Marhavý et al, 2019), *eto1-1* (Guzmán and Ecker, 1990); *wrky33-1* (SALK_006603), *mpk6-2* (SALK_073907), *glr3.3glr3.6* (Nguyen et al, 2018), *msl10-1* (Moe-Lange et al, 2021), *mca1mca2* (Yamanaka et al, 2010), and *acs* hextuple (*acs2-1, acs4-1, acs5-2, acs6-1, acs7-1, acs9-1*) (Mou et al, 2020). For transgenic lines or crossing lines, T3 or F3 generations, respectively, were used for our study.

Seeds of Arabidopsis were placed on ½ MS (Murashige and Skoog) agar plates. The seeds were stratified for 2 days at 4 °C. Seedlings were grown on vertically oriented plates in growth chambers at 150 μmol photons m$^{-2}$ s$^{-1}$, 22 °C under a 16-h light/8-h dark regime. Mutant and wild-type plants were grown in soil at 22 °C with a photoperiod of 16-h light and 8-h dark and 65% relative humidity. We use 5-day-old seedlings for our experiments of laser ablation and crushing roots.

## harmacological and hormonal treatments, and sample preparation

Five-day-old seedlings were transferred onto solid ½ MS media containing 0.9% agar with or without the indicated chemicals and incubated during imaging (for GdCl₃ and hormones, they were all pre-treated for at least 0.5 h, and kept continuously treated). The chemicals and hormones and their concentrations used in this study were as follows: gadolinium chloride (GdCl₃; 50 μM; Sigma-Aldrich/Merck), 1-aminocyclopropanecarboxylic acid (ACC; 1 μM; Sigma-Aldrich), aminoethoxyvinylglycine (AVG; 1 μM; Sigma-Aldrich/Merck), propidium iodide (PI; 15 mg/μl; Thermo Fisher), L-glutamate, (L-Glu, 100 μM; Sigma-Aldrich), adenosine triphosphate (ATP; 50 μM; Sigma-Aldrich), plant elicitor peptide (PEP1, 1 μM). To prepare for laser ablation and imaging, followed (Marhavý and Benkova, 2015; Marhavý et al, 2019).

To be more specific,

(1) prepare a sterilized chambered cover glass (Nunc™ Lab-Tek™ II Chambered Coverglass, Thermo Fisher), put 10 μl drops of propidium iodide (15 mg/μl) if needed on the cover glass and later the seedlings will mount on the cover glass on top of the drops to keep the seedlings in good condition not to be dehydrated.
(2) From the block of media cut off a 2×1×0.5 cm ½ MS media containing 0.9% agar medium with or without the indicated chemicals (for example in ACC experiment, the medium is supplied with 1 μM ACC) and submerged the medium into PI staining dye if needed.
(3) Transfer 10–15 seedlings inside the chamber, roots of individual seedlings must not overlap. Cover seedlings with the remaining block of media. Close with the chambered cover glass lid.

## Laser ablation and confocal imaging

Cell ablation experiments and imaging were performed with a STELLARIS 8 Multiphoton/Confocal Microscope from Leica Microsystems coupled with a Mai Tai Multiphoton laser. ROIs were drawn through cells prior to ablation. All the ablations were located at the beginning of the elongation zone of the roots.

To be more specific, in the experiment of quantifying the number of fluorescent nuclei

(1) Prepare STELLARIS 8 Multiphoton for use [set lasers (for GFP-488); objectives—40× glycerol immersion objective; image size —x: 242.28 μm, y: 120.94 μm, z: 19.00 μm; zoom - 1.2; scan mode—xyz; pixel dwell—1.02 μs; pinhole, 67.9 μm. laser power was set to 2% for *AOS::NLS-3xVenus* and *ACS6::NLS-3xVenus*; laser power was set to 4.5% for *JAZ10::NLS-3xVenus*. Fluorescence signals for yellow fluorescent protein (YFP) (excitation 515 nm, emission 525–560 nm) and propidium iodide (excitation 520 nm, emission 590 nm) were detected.

(2) Using FRAP (fluorescence recovery after photobleaching) mode, go to set up window, in this mode change the scan mode to xyt, change the MFP to SP667. The rest will be kept as in (1).

(3) Adding a new laser, (800 nm), turn on the MP1 shutter, keep the intensity as 0.

(4) Move to the FRAP mode, bleach window, setting the laser power 20–25%. Select the ROI (target region for ablation) in this window.

(5) Move to the FRAP mode, time course window, in this window, set up as 1 repetition, 2 for pre-bleach, 5 for bleach, and 2 for post-bleach.

(6) Move back to FRAP mode, change back to the Stellaris 8 mode for normal confocal imaging, and take the image immediately after ablation as time 0. Settings as (1).

(7) Imaging analysis: the Image J (NIH; http://rsb.info.nih.gov/ij) and LAX software packages were used. To quantify the number of fluorescent nuclei, open the image file in Image J, using Z project in the stack to setting max intensity for all images analysis.

## Ca$^{2+}$ wave velocity analyses upon laser ablation

Ca$^{2+}$ waves were monitored in neighboring undamaged cells next to the ablated cortex cells. The ablation method was followed as above.

(1) For imaging the Ca$^{2+}$ wave of *UBQ10pro::GCaMP3* using GFP setting (excitation 488 nm, emission 500–530 nm) and *R-GECO1* reporter lines, we used the values given in Keinath et al, 2015, (excitation 561 nm, emission 620–650 nm). The laser power was set to 4% for both of them. Images were recorded every 0.3 second (s) (Fig. 3; Appendix Fig. S6), for 3 s (10 images, Fig. 3; Appendix Fig. S6) or every 0.7 s for 7 s (10 images, Fig. 1; Appendix Fig. S7) prior to ablation for each sample; after ablation, every 0.3 s for 3 min (710 images for each sample; Fig. 3; Appendix Fig. S6); after ablation, every 0.7 s for 3 min (260 images for each sample; Fig. 1; Appendix Fig. S7).

(2) The fluorescence intensity of the ROI, (neighboring undamaged cell 100 μm from ablation cell), was measured in Image J. We measured 10 independent roots/images prior to ablation and took the average value as the 0-time point. We measured all the images after ablation (260 images for Fig. 1; Appendix Fig. S7, for each sample; 710 images for Fig. 3; Appendix Fig. S6, for each sample) then all the values were divided by the value we got for the 0-time point, to generate the relative value;

we consider this to represent relative fluorescence intensity. We used GraphPad Prism 10.4.0 to plot the Ca$^{2+}$ wave curve.

(3) To generate Movies EV1–EV4, Adobe Premiere Pro was used with 24 frames s$^{-1}$.

## Measurements of nematode-induced Ca$^{2+}$ dynamics

WT (Col-0) and *glr3.3glr3.6* plants stably expressing *proUBQ10::G-CaMP3* were raised on standard glass slides with agar medium supplemented with modified Knop's nutrient solutions (Sijmons et al, 1991). The growth conditions for the plants included a long-day cycle with 16 h of light followed by 8 h of darkness within a controlled growth chamber as previously described (Hasan et al, 2022). Cysts of *H. schachtii* were collected from a monoculture of mustard plants (Sinapis alba "Albatros").

(1) Cysts were incubated in a 3 mM ZnCl$_2$ solution to stimulate the hatching of juveniles. A freshly hatched cohort of 50–60 second-stage juveniles (J2s) was used to inoculate 10-day-old plants. Plants were left undisturbed for a few hours before being analyzed under a confocal microscope.

(2) Time-lapse imaging was conducted to capture the Ca$^{2+}$ wave induced by cyst nematode infection with a Leica SP8 lightning microscope (Leica Mikrosysteme) equipped with a dry objective (×10) and the Leica Application Suite X (LAS X) package. For the *GCaMP3* recordings, standard GFP settings were used (excitation at 488 nm, and emission at 515–520 nm), with the pinhole being adjusted to 1.

(3) The fluorescence intensity was measured in Image J, using the average of two ROIs, one being close to the point of infection and a ROI distal to the point of nematode entry. We quantified 5 independent root images before nematode infection and designated the average as the value for 0 s. The values obtained after nematode infection were normalized to the 0-second fluorescence intensity, thereby obtaining a relative fluorescence intensity.

(4) For statistical tests between the total WT fluorescence intensity and fluorescence obtained in the mutant, bar graphs were plotted with the total fluorescence along with one-way ANOVA and Tukey's Post-Hoc HSD comparisons at α = 0.05. Shapiro-Wilk's test was also performed on this dataset in order to confirm it's normal distribution.

## Nematode infection assays

Cyst nematode infection assays were carried out on Arabidopsis plants according to the protocol described by Chopra et al (2021).

(1) In each Petri dish, two Arabidopsis plants were grown on 0.2% (w/v) Knop medium. After 12 days of growth, the plants were inoculated with 60–80 freshly hatched, active *H. schachtii* juveniles (J2s) under sterile conditions.

(2) The assays were conducted across four independent biological replicates, with 20–30 plants of each genotype used per replicate. Fourteen days post-inoculation (dpi), the number of female and male nematodes per root system was recorded for evaluation.

## NemaWater treatment

NemaWater was prepared as described earlier (Mendy et al, 2017).

(1) briefly pre-infective second-stage juveniles of plant cyst nematode, *Heterodera schachtii* were incubated in water for 24 h at RT. Subsequently, the nematodes were removed, and the water was utilized for an overnight incubation of 5-day-old seedlings from the *AOS::NLS-3xVenus* line for 24 h. Meanwhile MilliQ Water ($H_2O$) treat the same group of plants for 24 h as control.

(2) For the ablation process, seedlings treated overnight were affixed to a chambered glass cover slide with ½ MS media block positioned over the roots. The treatment was sustained by adding 250 µl of *Hs*NemaWater and MilliQ Water ($H_2O$) onto the medium block covering the seedling roots. Perform laser ablation followed the same protocol as above. Confocal imaging was conducted immediately post-ablation and again after a 20-h interval following the ablation.

## RNA isolation, and quantitative real-time PCR (qRT-PCR)

Total RNA extraction was done using RNeasy Plant Mini Kit (QIAGEN) following the manufacturer introduction. Synthesis of complementary DNA was done using iScript cDNA Synthesis Kit (Bio-Rad Laboratories AB) and qRT-PCR using iTaq Univer SYBR Green SMX 200 (Bio-Rad Laboratories AB) following the manufacturer introduction. qRT-PCR was performed using an CFX96 Real-Time system (Bio-Rad). UBC1 (AT1G14400) served as reference gene for data analysis. Statistical significance was determined using ANOVA Tukey's multiple comparison test with a 95% confidence interval.

### Statistical analysis

All measurements or quantifications were from different samples and were repeated independently. All statistical analyses were performed using GraphPad Prism (10.4.0). We used ANOVA Tukey's multiple comparison test with a 95% confidence interval.

## Data availability

The microscopy data from this publication have been deposited to the BioImage Archive and assigned the accession number as S-BIAD1521. https://www.ebi.ac.uk/biostudies/bioimages/studies/S-BIAD1521.

The source data of this paper are collected in the following database record: biostudies:S-SCDT-10_1038-S44319-025-00471-z.

## Peer review information

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

## Acknowledgements

We thank Edward E Farmer and Vaughan Hurry for providing comments on our manuscript. We thank Adéla Lamaczová who helped in the lab with genotyping. We also acknowledge the facilities and technical assistance of the Umeå Plant Science Centre (UPSC) Microscopy Facility. Funding: This work was supported by funds from Vetenskapsrådet grant (2019-05634) (PM); Kempestiftelserna JCK-2011 (PM); Knut and Alice Wallenberg Foundation (KAW 2022.0029-Fate) (PM); MSA is supported by Marie Currie Postdoctoral

fellowship (MSCA PREENER 101066035); SM was supported by the DAAD (grant no. 57299294); Deutsche Forschungsgemeinschaft (DFG, German Research Foundation) supported UCV (INST 217/939-1 FUGG).

## Author contributions

**Xuemin Ma**: Conceptualization; Data curation; Formal analysis; Validation; Investigation; Visualization; Methodology; Writing—original draft; Writing—review and editing. **M Shamim Hasan**: Investigation; MSH conducted nematode infection experiment. **Muhammad Shahzad Anjam**: Investigation; MSA conducted HsNemaWater experiments. **Sakil Mahmud**: Investigation; SM conducted nematode infection experiment. **Sabarna Bhattacharyya**: Investigation; SB conducted nematode infection experiment. **Ute C Vothknecht**: Investigation; UCV conducted nematode infection experiment. **Badou Mendy**: Investigation; BM prepared 721 HsNemaWater. **Florian M W Grundler**: Investigation; FG conducted nematode infection experiment. **Peter Marhavý**: Conceptualization; Supervision; Funding acquisition; Methodology; Writing—original draft; Project administration; Writing—review and editing.

Source data underlying figure panels in this paper may have individual authorship assigned. Where available, figure panel/source data authorship is listed in the following database record: biostudies:S-SCDT-10_1038-S44319-025-00471-z.

## Funding

## Disclosure and competing interests statement

The authors declare no competing interests.

