## [Peer Review File · EMBO Reports]

Ca²⁺ Waves and Ethylene/JA Crosstalk Orchestrate Wound Responses in Arabidopsis Roots

Xuemin Ma, Shamim Hasan, Muhammad Anjam, Saki Mahmud, Sabarna Bhattacharyya, Ute Vothknecht, Badou Mendy, Florian Grundler, and Peter Marhavy

Corresponding author(s): Peter Marhavy (peter.marhavy@slu.se)

Review Timeline:

Transfer Date:	25th Oct 24
Editorial Decision:	14th Nov 24
Revision Received:	16th Dec 24
Editorial Decision:	18th Mar 25
Revision Received:	26th Mar 25
Editorial Decision:	31st Mar 25
Revision Received:	5th Apr 25
Accepted:	10th Apr 25

Transaction Report: This manuscript was transferred to EMBO reports following peer review at The EMBO Journal.

Referee #1:

The manuscript by Ma et al. elucidates a signaling pathway involved in local wounding, thus enhancing our understanding of the molecular network underlying plant responses to local damage. They uncovered the role of calcium wave propagation in regulating ethylene signaling and the feedback loop where ethylene signaling regulates calcium wave propagation. The authors have meticulously designed their experiments, revealing differences in the signaling mechanism between types of injury and cell types upon which injury is inflicted. Furthermore, it is commendable that the authors have backed their experiments with ample genetic analyses. I do have some suggestions and queries that would make the manuscript more easily accessible to readers. Here are my specific comments:

1- Line 87-89: "However, long-distance signals can also play a significant role in the wound response. Signals such as $[Ca^{2+}]_{cyt}$ (cytosolic free Ca^{2+}) and electrical signals can induce rapid leaflet movement in *Mimosa pudica* (Hagihara et al., 2022; Grenzi et al., 2023)." The authors claim long-distance signals play a role in wound response and then proceed to quote examples of leaflet movement in mimosa, which is responsive to even the slightest touch. Please provide examples where calcium (Annalisa Bellandi et al., Diffusion and bulk flow of amino acids mediate calcium waves in plants, Sci.) or electrical waves (Mousavi et al., 2013) are involved in wound response.

2- Line 119-12 and Sup Figure 1A: "The recording revealed an immediate increase in Ca^{2+} levels, peaking at 181 seconds for the local wave and 689 seconds for the distal wave." How far does the calcium wave propagate from the site of injury? Is ROI2 where the calcium wave terminates? If so, what would be the status of the calcium wave front between ROI1 and ROI2?

3- Line 132-138: "Subsequently, we investigated Ca^{2+} waves after single-cell ablation in *glr3.3glr3.6GCaMP3* (Nguyen et al., 2018), *msh10GCaMP3*, and *mca1mca2GCaMP3* lines." What is the regeneration phenotype in these mutants? Does perturbing the calcium waves impair regeneration?

4- Line 168-172: The WRKY33 expression seems to be higher in the vascular ablation compared to ablation of cortical cells in both the control and GdCl₃-treated lines. Could this be a cell type-specific response?

5- Line 175-188: WRKY33 is phosphorylated by MPK3/MPK6. From the data shown in the manuscript, there is some ACS6 expression in *wrky33* mutant, but not in *mpk6* mutant. When the authors claim, "Our results indicate that ACS6 expression is regulated by WRKY33 mediated by Ca^{2+} signals and also by MPK6," do they imply a WRKY33-independent pathway or a calcium signal-independent pathway (lines 148-153)?

6- Line 190-192: "Arabidopsis ovules transformed with *Physcomitrella patens* PpGLR1-dependent that ACC can activate cytosolic Ca²⁺ elevations by controlling GLR gating, resulting in transient cytosolic Ca²⁺ elevation." Dependent on what? Please rephrase.

7- Lines 216-218: "JA is well-described as a wound-associated hormone (Zhai et al., 2013; De Torres Zabala et al., 2016; Du et al., 2017). However, we previously showed that single-cell ablation in roots does not induce a strong JA response (Marhavý et al., 2019)." However, Zhou et al., 2019, Cell, shows that in vivo JA levels increase rapidly after laser ablation of stem cells. How do the authors reconcile this? Could it be a cell type-specific response?

8- Line 233-234: In Chang et al., 2013, EIN3 binding was found to modulate the ethylene signaling pathway, but direct binding of EIN3 to the AOS promoter is not shown in the main text of the manuscript. Can you confirm if you got this from the raw data analysis?

9- Line 241-249: The authors show that JAZ10::NLS-3xVenus was induced in the presence of an ethylene biosynthesis inhibitor and in mutants defective in the ethylene receptor, but reduced in ethylene mutants upon crushing. Please address this disparity.

10- Line 265-266: The authors claim to have revealed "a previously unrecognized relationship between nematode presence and the modulation of JA-responsive elements." But nematode infection had been shown to activate a JA-mediated regeneration pathway in roots in Zhou et al., 2019, Cell.

11- Line 190: Please correct the typo "show" to shown

12- Line 130: please correct the typo in "GCaMP3, an single-wavelength" to a single-wavelength.

Referee #2:

In this work by Ma and colleagues entitled "Integration of Ca²⁺ Signaling in Regulation of JA/Ethylene Pathways in Response to Local Damage in Arabidopsis Roots," the authors investigate the effects of cell laser ablation of root tissues. Specifically, they analyze cytosolic calcium dynamics in cells near the wounded area and the activation of promoters known to be activated by ethylene and jasmonic acid. This study is a continuation of previous work from the same author demonstrating that, unlike responses in leaves, where injury primarily induces JA synthesis in systemic leaves in roots, an accumulation of ethylene occurs.

This new piece of work confirms that only when the wounded cells are those of the cortex there is no activation of JA signaling, whereas, when the injury is made in vascular cells, JA signaling is activated, and in this latter case, ethylene plays a role as a negative regulator.

To investigate the possible involvement of calcium signaling upstream of this hormonal pathways activation, the authors assess the wound response in a series of mutants for genes encoding ion channels, previously shown to be involved in the propagation of electrical and calcium signals over long (i.e. GLRs and MSL10) and short distances (e.g. MCAs). Specifically, the authors selected the double mutant *glr3.3/glr3.6*, the single mutant *msl10*, and the double *mca1/mca2*. Analyses of calcium dynamics in root cells of these mutants upon laser ablation, show alterations in calcium dynamics, although not of significant magnitude. Nevertheless, in these mutants, promoters of genes responsive to ethylene and JA are not activated in response to injury. An indication regarding the possible role of Ca^{2+} upstream of this hormonal synthesis observed in response to injury is supported by genetic data where the mutant for the transcription factor CAMTA3 also shows a lack of ethylene-dependent promoter activation.

This work is sound, and well-designed, with solid genetic and cell biology approaches. The analyses are well executed, and the conclusions plausible, although, the role of Ca^{2+} as a second messenger in this process is possibly overestimated. As I report in the specific comments at various points in the text, the authors overstate the role of Ca^{2+} by not considering other factors, including the movement of other ions. I say this because the differences in calcium dynamics in the different mutants appear mild, despite the lack of promoter activation. This leads me to think that there may be more complex explanations.

- specific major concerns

Line 25. "...propagation mediated by...". The lack of *glrs*, *msl* or *mcas* only partially affects the cytosolic Ca^{2+} increase.

The maximum Ca^{2+} increase is almost the same with only a delayed peak. So, saying that the propagation is "mediated by..." seems an overstatement.

Line 119. "However, upon cyst nematode invasion of the roots, the recording revealed an immediate increase in Ca^{2+} levels, peaking at 181 seconds for the local wave and 689 seconds for the distal wave...".

A statistical analysis is required. I would also superimpose panels B and C to better

appreciate the propagation of Ca²⁺ signals and indicate the distance between the two analyzed ROIs.

Supplemental Figure 1D is not visible. Brightness and contrast need to be adjusted. I would also suggest using a different LUT, such as Fire. A false colour scale, for panels D-G must be added.

Line 128. "...selection mutants of three genes that encode Ca²⁺-permeable channels - GLRs, MSL10, and..."

As far as I know for the MSL10 there is no clear evidence that it is permeable to Ca²⁺, and it has been reported to have "preference" for anions.

Please, see Maksaev G, Haswell ES. MscS-Like10 is a stretch-activated ion channel from *Arabidopsis thaliana* with a preference for anions. *Proc Natl Acad Sci U S A*. 2012 Nov 13;109(46):19015-20. doi: 10.1073/pnas.1213931109.

Line 134. "...Interestingly, all these Ca²⁺ channel mutants exhibit a delay in the propagation of Ca²⁺ waves when compared to the wild type (WT) (Figure1B-1G, Movie S1 and S2), suggesting that Ca²⁺ plays a vital role in the regional wounding response, and specific Ca²⁺-permeable channels may modulate the extent of the calcium wave propagation..."

I have a general comment. Looking at the data reported in Figure 1 the Ca²⁺-wave phenotype is rather mild. Based on the results shown in Figure 1, I agree with authors that there is a difference among the genetic backgrounds but the phenotype is rather small, and the maximum Ca²⁺ increase is no different. So, theoretically, downstream Ca²⁺ sensors should still be activated but only with a mild delay. So, maybe the lack of downstream response might not be ascribed only to an altered Ca²⁺ signature.

Line 152. "Also, ACS6::NLS-3xVenus expression was significantly reduced after pronounced ablation of vascular regions (Supplemental Figure 2A and 2B) and mechanically crushing large cell populations..."

To be sure that the reporter is properly working in all genetic backgrounds it is important to show that the different lines show the same level of induction. Treatments with exogenous ethylene or the ACC precursor need to be performed.

Line 190. "It was show in Arabidopsis ovules transformed with *Physcomitrella patens* PpGLR1-dependent that ACC can activate cytosolic Ca²⁺ elevations by controlling GLR gating, resulting in transient cytosolic Ca²⁺ elevation (Mou et al., 2020)".

This is not correct. Mou et al. have expressed PpGLR1 in COS-7 mammalian cells that were treated with different amino acids and ACC (Figure 3). Independently they showed that Arabidopsis ovules treated with ACC showed a CNQX-sensitive cytosolic Ca²⁺ increase (Figure 4).

Line 200. In this paragraph, the authors show that ACC, ethylene biosynthesis or insensitive mutants show altered Ca²⁺ dynamics in response to laser ablation (Figure 3).

Since it was reported that ethylene activates a plasma membrane Ca²⁺-permeable channel in tobacco suspension cells (Zhao et al., 2007 *New Phytol*), I wonder if authors have tested the sole effect of ACC on plants in their experimental conditions.

GCaMP3 is an intensimetric indicator and the normalization of the data might lead to the loss of important information, such as a change in the calcium baseline, that cannot be precisely determined with an intensimetric sensor.

To evaluate the effect of the sole ACC on resting cytosolic Ca²⁺, I suggest performing experiments with plants expressing a ratiometric Ca²⁺ indicator. Either Nano YC65 Cameleon or mCherry-GCaMP6f Arabidopsis lines would perfectly fit with this request.

In the Figure legend, it would be useful to report how long the ACC pre-treatment was carried out.

- minor concerns

Line 49. "...Local tissue damage induces a rapid increase in cytosolic free Ca²⁺ levels, a process mediated by various Ca²⁺-permeable channel proteins...".

This is certainly true for the systemic responses, but locally the situation is different since these mutants still show a response. The *glr3.3/glr3.6* and *msl10* mutants show impaired

leaf-to-leaf propagation of electrical and calcium waves, and in particular for MSL10 Moe-Lange et al. report "...This study demonstrated that the mechanosensitive ion channel MSL10, present in plant vascular bundles, is required for component IV of wound-elicited electrical signals in distal leaves, as well as the amplitude and kinetics of the systemic Ca²⁺ wave...".

Line 88. "...and electrical signals can induce rapid leaflet movement in *Mimosa pudica* (Hagihara et al., 2022; Grenzi et al., 2023)..."

In Grenzi et al. the authors show the movement of the inflorescence stem in *Arabidopsis*.

Line 134. "...msl10GCaMP3, and mca1mca2GCaMP3 lines (Figure 1B-1G)..."

"GCaMP3" instead of "GCaMP3".

Line 134. "...Interestingly, all these Ca²⁺ channel mutants..."

Better to write: "...Interestingly, all these channel mutants..."

Line 186. "...WRKY33 mediated by Ca²⁺ signals and also by MPK6..."

For the same reason reported above, I would write: "...WRKY33 possibly mediated by Ca²⁺ signals and also by MPK6..."

In Figure 3, I strongly encourage authors to show the GCaMP3 fluorescence traces before the cell ablation...

Line 277. "To further confirm the regulation of JA responses by Ca²⁺ dependence, we conducted root crushing experiments..."

The channels under investigation are considered Ca²⁺-permeable channels and are not strictly specific to this ion. Since GLRs are permeable for Na⁺ (Wudick et al., 2018

Science), MSL channels are permeable to both anion and cation (Haswell et al., Current Biology 18, 730-734, May 20, 2008) and MCA are permeable to both anions and Ca²⁺ (i.e. MCA) (Yoshimura et al., Nat Commun 12, 6074 (2021)) I would better highlight the focus on the channel role rather than Ca²⁺ only. Indeed, based on results shown in Figure 1 these mutants show altered Ca²⁺ dynamics, but this might just tell us that their activity is affecting Ca²⁺ dynamics. So, just to be on the safe side I would suggest that Ca²⁺ is probably an important player, but without saying that is the only player.

Line 290. "Surprisingly, in the presence of GdCl₃, ein3eil1 did not exhibit any difference in AOS::NLS-3xVenus fluorescence before or after wounding, whereas AOS expression can be significantly reduced after wounding under WT conditions...".

Although it is statistically relevant the phenotype is quite mild.

Line 355. "In past decade number of studies have implicated the GLRs as the key Ca²⁺ regulators in systemic signal propagation (Mousavi et al., 2013; Toyota et al., 2018; Nguyen et al., 2018)...".

I would specify "in the shoot" since the role of GLRs in long-distance signaling has been primarily highlighted in the leaf vasculature.

In the Supplemental Movies, I would add the false colour scale.

In Figures 5A and F, a detailed description of box plots is missing.

"Supplemental Figure 10. JA response upon local wounding is dependent on Ca²⁺ signature".

I would say that it would be more appropriate to speak about the role of the selected channels. As I specified before, I think it is an oversimplification report that is only Ca²⁺ involved in this response.

L434. "...for Gdcl3..." "...for GdCl3...".

Line 439. "Prepare 2ml, 15 mg/ul PI in single well of 12-well-plate, then cut 2x1x0.5cm 1/2 MS media containing 0.9% agar medium with or without the indicated chemicals and submerged the medium into PI staining dye. Five-day-old seedlings were placed onto a chambered coverglass (Nunc{trade mark, serif} Lab-Tek{trade mark, serif} II Chambered Coverglass, ThermoFisher) with a 10µl drop of 15 mg/ul PI staining dye, covered with the medium pretreated with PI dye and immediately imaged during the relevant period (Marhavý and Benkova, 2015; Marhavý et al., 2019)".

This paragraph needs revision. It is written like a protocol and some ul needs to be changed with µl.

Referee #3:

This work is a continuation of (Marhavy et al 2019 EMBO J), and aims to characterise ionic and hormonal changes in Arabidopsis roots following nematode infection, single cell laser ablation (shown to serve as a proxy for early stages of nematode infection in Marhavy et al), and root crushing. The authors use several Ca²⁺ sensors, ethylene (ET) and jasmonate (JA) reporters coupled to mutants in respective pathways in various attempts to place the order of events occurring during the wound response, and the relative contribution of individual pathways. However, results are oftentimes conflicting among experiments and the interpretation of the data is not supported by the evidence provided due to the lack of complementary approaches. The information from specific experiments is interesting yet preliminary, and the overall manuscript lacks a coherent message on its main focus and on how this work advances the field. I agree with the authors' closing statement that the 'data provides insights into the intricate molecular mechanisms underlying plant responses to local damage induced in Arabidopsis roots'. Unfortunately, the data does not clarify specific relationships between Ca²⁺/ET/JA after root wounding nor nematode infections. Importantly, some of the methods are not described in sufficient detail, precluding the reproducibility of the results shown.

MAJOR CONCERNS

-The authors claim that:

1. the ablation-induced upregulation of WRKY33/MPK66 (known to act upstream of ET biosynthesis genes) and ACO1/ACS6 (ET biosynthesis genes) is reduced in Ca²⁺ mutants (glrs, msl10, mcas), indicating that Ca²⁺ ions are upstream of ET biosynthesis (which has already been shown in Marhavy et al)

2. The ET pathway inhibits JA responses after wounding, as the activity of JA marker genes is enhanced in ET mutant backgrounds and impaired by external ET (ACC) applications (Fig.4). At the same time, the induction of JA marker genes is reduced (not enhanced) in other mutants of the ET pathway (Suppl Fig. 8B), showing strong inconsistencies between claims and data presented.

3. Laser ablation in the vasculature (but not in cortex cells) induces JA marker genes in a Ca²⁺ dependent manner

All the evidence on gene expression is based exclusively on quantifying the number of fluorescent nuclei of promoter fusions driving the expression of NLS-3xVENUS of ET- and JA-responsive promoters. In other words, the manuscript bases all the conclusions on promoter activities of transcriptional reporters and does not provide any evidence on actual transcript levels. Furthermore, interpretations from ET and JA reporter lines is particularly concerning due to the lack of a common normalizer among genotypes and the risk of silencing or selection of transgenic lines. Also, the methods do not report how were the Ca²⁺/ET/JA reporters introduced in different mutant backgrounds, i.e. by crossing or transformation. Transcript levels of the transgenes should be evaluated across genetic backgrounds or alternative reporters used (ET and JA biosensors are available), and results should be verified with complementary approaches (eg. qPCR, etc). In the current form, the conclusions are simply not sound enough.

- The ET pathway affects the Ca²⁺ wave after laser ablation, in a putative feedback regulation (Fig. 3). Conclusions are based on external ET (ACC) applications and Ca²⁺ imaging in ET mutant lines. It remains unclear at what timepoint after wounding does a feedback mechanism become relevant. As is, the message is confusing: Ca²⁺ is both upstream and downstream of the ET pathway.

- Are Ca²⁺ permeable channels (eg. glrs, msl10, mcas), ET and JA loss of function mutants affected in plant responses to nematodes? In other words, given that the molecular findings are not recapitulated by NemaWater treatment (Fig.4), when do the proposed molecular mechanisms become functionally relevant?

- Student's t-test is not appropriate to evaluate differences among more than 2 groups (whole manuscript)

OTHER CONCERNS

- GLR3.3 & 3.6 impact both electrical and Ca²⁺ signals in leaves. How do the authors reconcile their current findings on reduced Ca²⁺ signals in *glr3.3/3.6* (Fig. 1) with their earlier findings on *glr3.3/3.6* not affecting electrical signals in roots? Are electrical signals disengaged from changes in extracellular Ca²⁺ only in roots?
- The introduction section, L.78-87, contains many inaccuracies. Eg1. JA-deficient mutants propagate wounding-induced long-distance signals similar to WT plants (i.e JA is not required for systemic signal propagation after wounding). Eg2. The closure of plasmodesmata has been associated to hampering the spread of pathogens and harmful molecules, rather than limiting the spread of wound signals
- what is the rationale of using different Ca²⁺ reporters between experiments (R-GECO1 in some cases vs GCaMP3 in others)?
- Fig.1 While the quantification of reporter intensity following ablation was performed only in the shootward direction, the images show a bidirectional signal propagation. How does the speed of propagation compare to what is known in leaves?
- Lines 145-146, gene names should be in capital italics
- Lines 150-156. Relative fluorescence intensity (text) is not the same as counting the frequency of fluorescent nuclei (Fig.2), rendering the text and the data presented confusing and ambiguous
- Suppl Fig7. There is no panel for C, nor any quantification

Dear Peter,

Thank you for the transfer of your research manuscript to EMBO Reports. Please accept my apologies regarding the delay in handling your manuscript.

As William indicated, we are interested in potential publication of your study at EMBO Reports. From the informal feedback William received on your rebuttal, we think that the study could be published with appropriate toning down of causality and the Ca²⁺ dependent role of the studied ion channels.

You have now transferred the original version that had been reviewed at The EMBO Journal. In order for you to update the files, submit the revised version of your manuscript and the point-by-point response to the referee concerns, I am now making a decision on your manuscript, inviting you to revise it for EMBO Reports. Once the updated files have been uploaded, I will contact the former referees again, asking them to re-review your study and submit formal reports to complement the informal feedback we had received.

Please find below the formatting guidelines for EMBO Reports. Since your revised manuscript is ready, I list the most important points here:

- The figures must be uploaded as one file per figure. The figure legends are part of the manuscript text and should be removed from the figure files (point 2 below).
- Supplemental Information is called "Appendix". This is a separate PDF file that includes a table of content with page numbers. The nomenclature is Appendix Figure S#. (point 6)
- We need an Author Checklist (point 4).
- The Data and Materials availability is called "Data Availability" and needs to go to the end of the Methods. Please rephrase to "This study includes no data deposited in external repositories."
- The Author Contributions should be removed and only be specified in the online manuscript tracking system. They are retrieved from there and automatically typeset into the article upon publication.
- Make sure all information on funding is specified in the system.
- We will need a Reagents and Tools table (Point 12), either now or at final revision stage.
- You will be contacted and asked to supply all source data. It would be great if we had these data already now, but please let me know in case you would need more time to complete this.

Kind regards,

Martina

=====

Formatting guidelines:

2) individual production quality figure files as .eps, .tif, .jpg (one file per figure). Please download our Figure Preparation Guidelines (figure preparation pdf) from our Author Guidelines pages <https://www.embopress.org/page/journal/14693178/authorguide> for more info on how to prepare your figures.

4) a complete author checklist, which you can download from our author guidelines (<<https://www.embopress.org/page/journal/14693178/authorguide>>). Please insert information in the checklist that is also reflected in the manuscript. The completed author checklist will also be part of the RPF.

5) Please note that all corresponding authors are required to supply an ORCID ID for their name upon submission of a revised manuscript (<<https://orcid.org/>>). Please find instructions on how to link your ORCID ID to your account in our manuscript tracking system in our Author guidelines (<<https://www.embopress.org/page/journal/14693178/authorguide#authorshipguidelines>>)

6) We replaced Supplementary Information with Expanded View (EV) Figures and Tables that are collapsible/expandable online. A maximum of 5 EV Figures can be typeset. EV Figures should be cited as 'Figure EV1, Figure EV2' etc... in the text and their respective legends should be included in the main text after the legends of regular figures.

7) Please include a dedicated "Data Availability" section at the end of the Methods (suggested wording: "The [structural coordinates | microarray | mass spectrometry] data from this publication have been deposited to the [name of the database] database [URL] and assigned the identifier [accession | permalink | hashtag]."). Should this not apply, this should still be stated as "This study includes no data deposited in external repositories."

Additional information on source data and instruction on how to label the files are available <<https://www.embopress.org/page/journal/14693178/authorguide#sourcedata>>.

10) Figure legends and data quantification:
The following points must be specified in each figure legend:

- the name of the statistical test used to generate error bars and P values,
 - the number (n) of independent experiments (please specify technical or biological replicates) underlying each data point,
 - the nature of the bars and error bars (s.d., s.e.m.)
- If the data are obtained from n {less than or equal to} 5, show the individual data points in addition to the SD or SEM.
- If the data are obtained from n {less than or equal to} 2, use scatter blots showing the individual data points.

See also the guidelines for figure legend preparation:
<https://www.embopress.org/page/journal/14693178/authorguide#figureformat>

11) Our journal encourages inclusion of *data citations in the reference list* to directly cite datasets that were re-used and obtained from public databases. Data citations in the article text are distinct from normal bibliographical citations and should directly link to the database records from which the data can be accessed. In the main text, data citations are formatted as follows: "Data ref: Smith et al, 2001" or "Data ref: NCBI Sequence Read Archive PRJNA342805, 2017". In the Reference list, data citations must be labeled with "[DATASET]". A data reference must provide the database name, accession number/identifiers and a resolvable link to the landing page from which the data can be accessed at the end of the reference. Further instructions are available at <<https://www.embopress.org/page/journal/14693178/authorguide#referencesformat>>.

12) All Materials and Methods need to be described in the main text using our 'Structured Methods' format. According to this format, the Methods section includes a Reagents and Tools Table (listing key reagents, experimental models, software and relevant equipment and including their sources and relevant identifiers) followed by a Methods and Protocols section describing the methods, ideally using a step-by-step protocol format. The aim is to facilitate adoption of the methodologies across labs. Please download and fill our Reagents and Tools Table template (.docx), which you can find in our author guidelines:

13) As part of the EMBO publication's Transparent Editorial Process, EMBO Reports publishes online a Review Process File to accompany accepted manuscripts. This File will be published in conjunction with your paper and will include the referee reports, your point-by-point response and all pertinent correspondence relating to the manuscript.

=====

Dear Editor,

We would like to thank the reviewers for their valuable comments that allowed us to significantly improve our manuscript "*Regional Ca²⁺ Wave, Hormone Responses upon Laser Ablation in Arabidopsis*". We have amended the current manuscript with new data, which further strengthen our work and support the conclusions. Please find our point-by-point response to the reviewer comments below. The response to reviewers' comments are highlighted in blue and changes in the manuscript are highlighted in red.

Referee #1:

The manuscript by Ma et al. elucidates a signaling pathway involved in local wounding, thus enhancing our understanding of the molecular network underlying plant responses to local damage. They uncovered the role of Ca²⁺ wave propagation in regulating ethylene signaling and the feedback loop where ethylene signaling regulates Ca²⁺ wave propagation. The authors have meticulously designed their experiments, revealing differences in the signaling mechanism between types of injury and cell types upon which injury is inflicted. Furthermore, it is commendable that the authors have backed their experiments with ample genetic analyses. I do have some suggestions and queries that would make the manuscript more easily accessible to readers. Here are my specific comments:

We would like to thank Reviewer 1 for his/her suggestions to improve our manuscript!

1- Line 87-89: "However, long-distance signals can also play a significant role in the wound response. Signals such as [Ca²⁺]cyt (cytosolic free Ca²⁺) and electrical signals can induce rapid leaflet movement in *Mimosa pudica* (Hagihara et al., 2022; Grenzi et al., 2023)." The authors claim long-distance signals play a role in wound response and then proceed to quote examples of leaflet movement in mimosa, which is responsive to even the slightest touch. Please provide examples where calcium (Annalisa Bellandi et al., Diffusion and bulk flow of amino acids mediate Ca²⁺ waves in plants, Sci.) or electrical waves (Mousavi et al., 2013) are involved in wound response.

We thank to the reviewer for the valuable suggestions. Now we have implemented the recommended changes into the text.

2- Line 119-12 and Sup Figure 1A: "The recording revealed an immediate increase in Ca²⁺ levels, peaking at 181 seconds for the local wave and 689 seconds for the distal wave." How far does the Ca²⁺ wave propagate from the site of injury? Is ROI2 where the Ca²⁺ wave terminates? If so, what would be the status of the Ca²⁺ wave front between ROI1 and ROI2?

The figure is now replaced by data using GCaMP3 instead of the R-GECO1 lines. ROI2 is not where the Ca²⁺ wave terminates.

We thank the reviewer for raising these insightful questions. As mentioned in the methods section, nematode infection is an unpredictable natural process that requires extended periods of focused observation to capture a single event. Therefore, numerous attempts were made to track and record a successful root invasion by cyst nematode juveniles. The results represent

the first observation that nematode invasion can trigger Ca^{2+} bursts. Given the complexity of the infection process, a more in-depth and detailed study is needed to fully understand Ca^{2+} wave propagation from the point of injury and the dynamics of signaling in both local and distal areas of infection, which we plan to investigate in a follow-up study

In response to Reviewer 2's suggestions, we have replaced the R-GECO1 line with GCaMP3 and used both WT and the *glr3.3glr3.6* mutant line to maintain consistency with the other experiments. This figure highlights the differences in Ca^{2+} dynamics between wild-type (WT) and *glr3.3glr3.6* mutant plants during nematode infection. As shown, the absence of AtGLR3.3/GLR3.6 completely abolished the nematode-induced Ca^{2+} spike. We used two regions of interest (ROIs)-local (ROI1) and distal (ROI2)-which were combined here to quantify relative intensity. However, ROI2 does not represent the termination of the Ca^{2+} wave.

3- Line 132-138: "Subsequently, we investigated Ca^{2+} waves after single-cell ablation in *glr3.3glr3.6GCaMP3* (Nguyen et al., 2018), *msl10GCaMP3*, and *mca1mca2GCaMP3* lines." What is the regeneration phenotype in these mutants? Does perturbing the Ca^{2+} waves impair regeneration?

Although we did not investigate the regeneration process in these three mutants, it is indeed an intriguing area of study. Hernández-Coronado et al. (2022) in *Developmental Cell* demonstrated that GLRs play a significant role in the regeneration process via the salicylic acid signaling pathway. Their findings revealed that the mutants *glr3.1/glr3.2/glr3.3/glr3.6* exhibited enhanced regeneration efficiency following the excision of the root cap. These results suggest a potential connection between GLRs function and improved regenerative capacity, warranting further exploration, which is beyond the scope of this manuscript.

4- Line 168-172: The WRKY33 expression seems to be higher in the vascular ablation compared to ablation of cortical cells in both the control and GdCl₃-treated lines. Could this be a cell type-specific response?

Thanks to Reviewer 1 for pointing this out. As indicated, the *WRKY33::NLS-YFP* signal is indeed more pronounced in the vascular region under the ablation condition. This suggests that WRKY33 may have a cell type-specific response to laser ablation, with a stronger activation in vascular cells compared to cortical cells. This finding could point to the possibility that WRKY33 is more actively involved in the stress response within vascular tissues.

(A, B) Quantification of maximum projection images XYZ of *WRKY33::YFP-NLS* in WT (Col-0) before (0h) and 5 hours after laser ablation (5h) in the cortex cells (cor), in the vascular region (vas) with (B) or without (A) treatment with 50 μ M GdCl₃. The graph shows a number of cells with positive nuclear (YFP-NLS) signals. N = three biological pools, each pool includes 5 seedlings (ANOVA Tukey's multiple comparison test with a 95% confidence interval).

5- Line 175-188: WRKY33 is phosphorylated by MPK3/MPK6. From the data shown in the manuscript, there is some ACS6 expression in *wrky33* mutant, but not in *mpk6* mutant. When the authors claim, "Our results indicate that ACS6 expression is regulated by WRKY33 mediated by Ca²⁺ signals and also by MPK6," do they imply a WRKY33-independent pathway or a calcium signal-independent pathway (lines 148-153)?

Our results indicate that the *ACS6* response to local wounding is primarily regulated by WRKY33, as evidenced by the reduced expression of *ACS6* in the *wrky33-1* mutant (Figure 2B, 2C, and Appendix Figure S3). This is further supported by previous findings showing that WRKY33 directly binds to the W-box motif in the *ACS6* promoter (Li et al., 2012). Therefore, in our study, *ACS6* is directly and positively regulated by WRKY33.

Our data (Figure 2D, 2E) indicate that GdCl₃, an inhibitor of calcium channels, suppresses *WRKY33* expression. Furthermore, evidence from Appendix figure S4D shows that *WRKY33* expression is reduced in the *glr3.3glr3.6*, *mssl10-1*, and *mca1mca2* mutants, where the Ca²⁺ wave is altered following local wounding. This reduction corresponds with a repression of *ACS6* expression. These findings imply that GLR3.3/GLR3.6, MSL10, and MCA1/MCA2 might participate in Ca²⁺-mediated processes potentially associated with the *WRKY33-ACS6* regulatory pathway.

MPK6 is also a key regulator of *ACS6*, influencing both its transcription and protein stability (Li et al., 2012). In our study, *ACS6* expression is diminished in the *mpk6-2* mutant, underscoring the significance of MPK6 in *ACS6* regulation. While there is evidence hinting at an indirect connection between GLR3.6 and MPK6, particularly in tomato studies, the direct relationship between MPK6 and GLR3.3/GLR3.6 in Arabidopsis has yet to be established. For instance, root-knot nematode *Meloidogyne incognita* activates mitogen-activated protein kinases SIMPK1 and SIMPK2 (tomato orthologs of Arabidopsis MPK6) (Nie et al., 2013), and this activation is attenuated in *slglr3.5* grafted tomato plants (where GLR3.5 is homologous to Arabidopsis GLR3.3/GLR3.6) (Wang et al., 2019; Shao et al., 2020). Similarly, injection-induced MPK1/2 activity is reduced in *glr3.5* mutant tomato leaves (Wang et al., 2019).

Although direct evidence connecting GLR3.3/GLR3.6 to MPK6 in Arabidopsis is limited, these observations suggest a possible involvement of GLR3.3/GLR3.6 in wounding-related MPK signaling cascades, which could influence downstream gene regulation.

Given the complexity of these signaling pathways, it is possible that there could be a WRKY33-independent pathway, such as a Ca²⁺-MPK6-*ACS6*, regulating *ACS6*. Furthermore, *ACS6* could potentially be targeted by other transcription factors, which yet they need to be uncovered in future.

6- Line 190-192: "Arabidopsis ovules transformed with *Physcomitrella patens* PpGLR1-dependent that ACC can activate cytosolic Ca²⁺ elevations by controlling GLR gating, resulting in transient cytosolic Ca²⁺ elevation." Dependent on what? Please rephrase.

Now we have rephrased the sentence. The sentence refers to an experiment where the activation efficiency of different ligands for GLR was tested using COS-7 mammalian cells expressing *Physcomitrella patens* GLR1 (*PpGLR1*) along with a Ca²⁺ sensor. In this setup, ACC was found to be the most efficient ligand in inducing calcium influx through *PpGLR1* when exogenous Ca²⁺ was present ((Mou et al., 2020). Additionally, ACC was shown to promote an enhanced calcium influx in the Arabidopsis ovule as well (Mou et al., 2020). Therefore, the ACC-dependent activation refers to ACC's ability to efficiently gate *PpGLR1*, leading to a transient elevation in cytosolic Ca²⁺.

7- Lines 216-218: "JA is well-described as a wound-associated hormone (Zhai et al., 2013; De Torres Zabala et al., 2016; Du et al., 2017). However, we previously showed that single-cell ablation in roots does not induce a strong JA response (Marhavý et al., 2019)." However, Zhou et al., 2019, Cell, shows that in vivo JA levels increase rapidly after laser ablation of stem cells. How do the authors reconcile this? Could it be a cell type-specific response?

Thank you for pointing out this important detail. We have revised the statement. The key message here is that laser ablation in cortex cells does not trigger a JA response in at least three JA reporter lines, please see provided data below. However, our data indicate that two of these reporter lines did show an enhanced response when the ablation occurred in the vascular region. Additionally, we observed that *JAZ10* responded to QC cell ablation (see data below). Therefore, our findings suggest that JA responds to laser ablation in a cell type-specific manner, rather than uniformly across all cell types.

(A) Quantification of maximum projection images XYZ of *JAZ10::NLS-3xVenus* (*JAZ10*), *AOS::NLS-3xVenus* (*AOS*), and *MYC2::NLS-3xVenus* (*MYC2*), the jasmonate-response marker line in WT (Col-0) before (non-LA) and 5 hours after laser ablation (LA) in the cortex cells in a number of cells with a positive nuclear (NLS-3xVenus) signal. (B) Quantification of maximum projection images XYZ of *JAZ10::NLS-3xVenus* (*JAZ10*), *AOS::NLS-3xVenus* (*AOS*), and *MYC2::NLS-3xVenus* (*MYC2*), the jasmonate-response marker line in WT (Col-0) before (non-LA) and 5 hours after laser ablation (LA) in the vascular region in a number of cells with a positive nuclear (NLS-3xVenus) signal. (C) Quantification and representative of the (XYZ) maximum projection images XYZ of *JAZ10::NLS-3xVenus* in WT (Col-0), before (non-LA) and after 5 hours laser

ablation (LA) in QC (quiescent center) cell, in a number of cells with positive nuclear (NLS-3xVenus) signal. In (A), (B) and (C), N = three biological pools, each pool includes 5 seedlings (Student's *t*-test; ***P* < 0.01, and ****P* < 0.0001, ns: not statistically significant).

8- Line 233-234: In Chang et al., 2013, EIN3 binding was found to modulate the ethylene signaling pathway, but direct binding of EIN3 to the AOS promoter is not shown in the main text of the manuscript. Can you confirm if you got this from the raw data analysis?

The data were extracted from the supplemental excel, name elife-00675-suppl1-v1. Sheet B,

EIN3 targets	
EIN3-R (EIN3 targets, ethylene-regulated)	
AtID	TAIR10 Primary Gene Symbol/Gene Model Description
AT5G42650	ALLENE OXIDE SYNTHASE (AOS)

9-nLine 241-249: The authors show that JAZ10::NLS-3xVenus was induced in the presence of an ethylene biosynthesis inhibitor and in mutants defective in the ethylene receptor, but reduced in ethylene mutants upon crushing. Please address this disparity.

The observed disparity can be explained by the complex interplay between ethylene and JA signaling pathways, as well as the differences in ethylene sensitivity between mutants.

In our study, we observed that in the *etr1-1* mutant, which completely lacks ethylene binding and exhibits a strong ethylene-insensitive phenotype, the *JAZ10::NLS-3xVenus* reporter line showed a more pronounced response in the local damaged region (our data). This is consistent with the findings of Rojo et al. (1999), where the *etr1-3* mutant, which retains some sensitivity to ethylene, also showed increased expression of JA-responsive genes such as *VSP*, *JR1*, and *JR2* in response to wounding. The *etr1-1* completely removed ethylene binding which yields very strong ethylene-insensitive phenotype, while the *etr1-3* plants still have some sensitivity to ethylene, but they both affect ethylene perception and transduction (Hall et al., 1999).

Additionally, in the *acs* octuple mutant studied by Tsuchisaka et al. (2009), *JAZ10* was identified among the induced genes (in their supplemental table S4), indicating that the absence of ethylene can lead to enhanced JA signaling. Conversely, in the *eto1-1* mutant, where ethylene is overproduced due to the negative regulation of ACS by *ETO1*, both *JAZ10* and *AOS* were strongly repressed upon local wounding by crushing roots - the *ein2-1* which has more pronounced ethylene production at seedling stage (5day-old) under normal condition (Woeste et al., 1999), and showed reduced *AOS* (in the graph below showing the qRT-PCR result of *AOS* expression in *eto1-1* local wounding condition and *JAZ10* expression compared to the WT (Appendix Figure S9A, S9B, S10D, S10E).

Further supporting our findings, in the *ein3eill* mutant, we observed increased *AOS* expression following local wounding, as evidenced by the *AOS::NLS-3xVenus* reporter line. This aligns with previous research showing that EIN3 directly binds to the *AOS* promoter

(Chang et al., 2013). Interestingly, despite EIN3/EIL1 known interactions with JAZ1, JAZ3, and JAZ9 at the protein level (Zhu et al., 2011), our data show that JAZ10 expression is repressed in the *ein3eil1* mutant upon wounding. This suggests that EIN3/EIL1 may positively regulate *JAZ10* in a context-dependent manner.

In summary, the differences in *JAZ10* expression across various ethylene mutants likely reflect the nuanced and context-specific nature of JA-ethylene crosstalk, where individual genes may behave differently depending on the specific signaling environment.

(A)

REDACTED:
Supplemental Table S4, Tsuchisaka et al., 2009

(B)

(A) The data were extracted from the supplemental Table S4, Tsuchisaka et al., 2009. (B) Expression of *AOS* was determined by qRT-PCR in WT (Col-0), *eto1-1* with (wounding) or without wounding (control). Samples were harvested 5h after wounding. Data are means of three biological replicates, ANOVA Tukey's multiple comparison test was used with a 95% confidence interval.

10- Line 265-266: The authors claim to have revealed "a previously unrecognized relationship between nematode presence and the modulation of JA-responsive elements." But nematode infection had been shown to activate a JA-mediated regeneration pathway in roots in Zhou et al., 2019, Cell.

To maintain accuracy, we have decided to remove the sentence. Zhou et al. (2019), which focused on root-knot nematode *Meloidogyne incognita*. Now we emphasize our findings and ensures in the discussion to accurately reflects the specific context of our study.

11- Line 190: Please correct the typo "show" to shown

This was Corrected

12- Line 130: please correct the typo in "GCaMP3, an single-wavelength" to a single-wavelength.

This was Corrected

Referee #2:

In this work by Ma and colleagues entitled "Integration of Ca²⁺ Signaling in Regulation of JA/Ethylene Pathways in Response to Local Damage in Arabidopsis Roots," the authors investigate the effects of cell laser ablation of root tissues. Specifically, they analyze cytosolic calcium dynamics in cells near the wounded area and the activation of promoters known to be activated by ethylene and jasmonic acid. This study is a continuation of previous work from the same author demonstrating that, unlike responses in leaves, where injury primarily induces JA synthesis in systemic leaves in roots, an accumulation of ethylene occurs.

This new piece of work confirms that only when the wounded cells are those of the cortex there is no activation of JA signaling, whereas, when the injury is made in vascular cells, JA signaling is activated, and in this latter case, ethylene plays a role as a negative regulator.

To investigate the possible involvement of calcium signaling upstream of this hormonal pathways activation, the authors assess the wound response in a series of mutants for genes encoding ion channels, previously shown to be involved in the propagation of electrical and calcium signals over long (i.e. GLRs and MSL10) and short distances (e.g. MCAs). Specifically, the authors selected the double mutant *glr3.3/glr3.6*, the single mutant *msl10*, and the double *mca1/mca2*. Analyses of calcium dynamics in root cells of these mutants upon laser ablation, show alterations in calcium dynamics, although not of significant magnitude. Nevertheless, in these mutants, promoters of genes responsive to ethylene and JA are not activated in response to injury. An indication regarding the possible role of Ca²⁺ upstream of this hormonal synthesis observed in response to injury is supported by genetic data where the mutant for the transcription factor CAMTA3 also shows a lack of ethylene-dependent promoter activation.

This work is sound, and well-designed, with solid genetic and cell biology approaches. The analyses are well executed, and the conclusions plausible, although, the role of Ca²⁺ as a second messenger in this process is possibly overestimated. As I report in the specific comments at various points in the text, the authors overstate the role of Ca²⁺ by not considering other factors, including the movement of other ions. I say this because the differences in calcium dynamics in the different mutants appear mild, despite the lack of promoter activation. This leads me to think that there may be more complex explanations.

- specific major concerns

Line 25. "...propagation mediated by...". The lack of *glrs*, *msl* or *mcas* only partially affects the cytosolic Ca²⁺ increase.

The maximum Ca²⁺ increase is almost the same with only a delayed peak. So, saying that the propagation is "mediated by..." seems an overstatement.

We would like to thank Reviewer 2 for his/her positive and encouraging comments!

We revised the statement in both the conclusion and the results section. The updated conclusion now reads:

“Interestingly, all these three mutants exhibit a retardation in the propagation of Ca^{2+} waves when compared to the wild type (WT) upon laser ablation (Figure 1B–1G, Movie S1 and S2), suggesting that Ca^{2+} plays a vital role in the regional wounding response, and specific Ca^{2+} -permeable channels GLR3.3/GLR3.6 and MCA1/MCA2, may modulate the extent of the Ca^{2+} wave propagation. MSL10 is also important for Ca^{2+} wave propagation upon laser ablation.”

In the discussion, we have incorporated the following revised statement:

“Also, it supports that MSL10 functions in the same pathway with GLRs (Moe-Lange et al., 2021). MSL10 has been shown to involve in a transient calcium influx in response to cell swelling (Basu and Haswell, 2020). MCA1/MCA2 were shown to mediate Ca^{2+} uptake in plants, and they are responsible for the calcium influx in yeast (Nakagawa et al., 2007; Yamanaka et al., 2010). Our data showed that MCA1/MCA2, also participates in Ca^{2+} wave propagation upon laser ablation, and together with the result of GLR3.3/GLR3.6 and MSL10, it indicates that the immediate Ca^{2+} levels elevation in the undamaged cell may depend on multiple players.”

Line 119. "However, upon cyst nematode invasion of the roots, the recording revealed an immediate increase in Ca^{2+} levels, peaking at 181 seconds for the local wave and 689 seconds for the distal wave...".

A statistical analysis is required. I would also superimpose panels B and C to better appreciate the propagation of Ca^{2+} signals and indicate the distance between the two analyzed ROIs.

Supplemental Figure 1D is not visible. Brightness and contrast need to be adjusted. I would also suggest using a different LUT, such as Fire. A false color scale, for panels D-G must be added.

As explained above, we have now used GCaMP3 reporter lines. A statistical analysis with the quantified total fluorescence intensity over the various time points in between the WT and the double mutant (*glr3.3glr3.6*) has now been made, along with an ANOVA test which clearly shows the difference (a clear reduction in the Ca^{2+} wave intensity in the double mutant). Please refer to the figure below. In the revised figure, the fluorescence intensity in the current microscopic images (Appendix Figure 1) is much clearer, and we believe there is no need for a false color scale.

Quantification of the absolute fluorescence intensities upon nematode infection in the roots of the WT and *glr3.3glr3.6* mutant. For the calculation, values from the ROIs were added cumulatively over the total period of measurements. Data represents means \pm SE of five independent replicates. Statistical significances were estimated with the help of one-way ANOVA and Tukey's HSD tests.

Line 128. "...selection mutants of three genes that encode Ca²⁺-permeable channels - GLRs, MSL10, and..."

As far as I know for the MSL10 there is no clear evidence that it is permeable to Ca²⁺, and it has been reported to have "preference" for anions.

Please, see Makshev G, Haswell ES. MscS-Like10 is a stretch-activated ion channel from *Arabidopsis thaliana* with a preference for anions. *Proc Natl Acad Sci U S A*. 2012 Nov 13;109(46):19015-20. doi: 10.1073/pnas.1213931109.

Thank you for bringing this to our attention. We have corrected the description in the text to accurately reflect the role of MSL10. In the updated version of our manuscript we specified the individual functions of GLRs, MSL10, and MCA1/MCA2 in the context of our data analysis, rather than grouping them together, to ensure greater accuracy in our interpretation.

Line 134. "...Interestingly, all these Ca²⁺ channel mutants exhibit a delay in the propagation of Ca²⁺ waves when compared to the wild type (WT) (Figure1B-1G, Movie S1 and S2), suggesting that Ca²⁺ plays a vital role in the regional wounding response, and specific Ca²⁺-permeable channels may modulate the extent of the Ca²⁺ wave propagation..."

I have a general comment. Looking at the data reported in Figure 1 the Ca²⁺-wave phenotype is rather mild. Based on the results shown in Figure 1, I agree with authors that there is a difference among the genetic backgrounds but the phenotype is rather small, and the maximum Ca²⁺ increase is no different. So, theoretically, downstream Ca²⁺ sensors should still be activated but only with a mild delay. So, maybe the lack of downstream response might not be ascribed only to an altered Ca²⁺ signature.

Upon laser ablation, which is a strong and controlled method to induce calcium influx, we anticipated a significant Ca²⁺ response in adjacent cells. However, the observed Ca²⁺ wave phenotype in the mutants, though present, appears relatively mild compared to the wild type. We speculate that the minimal differences in Ca²⁺ wave propagation could be due to several factors:

1. ROI Distance: The region of interest (ROI) we measure is quite small relative to the site of damage, which might limit the observed differences in Ca²⁺ wave propagation.
2. Complexity of Laser Ablation: The laser ablation method is designed to induce a strong Ca²⁺ influx, but this context is complex and influenced by multiple factors, including potential diffusion of extracellular ATP (eATP) and ATP signaling. Studies like Donati et al. (2021) have shown that eATP diffusion is crucial for Ca²⁺ wave propagation, suggesting that additional factors beyond just Ca²⁺ influx are involved.

Regarding downstream responses, we currently lack an ideal method to directly monitor changes in calcium signatures and their impact on downstream signaling. However, our indirect evidence supports a role for GLR3.3/GLR3.6, MSL10, and MCA1/MCA2 in signaling cascades and hormone responses. For example, the study by Marhavý et al. (2019) showed that GdCl₃, a calcium channel inhibitor, attenuates the response of the *ACS6::NLS-3xVenus* ethylene reporter line upon laser ablation, suggesting that calcium channels may also play a role in downstream signaling.

Additionally, *glr3.3glr3.6* mutants have shown increased susceptibility to cyst nematode infection (Appendix Figure S5), while ACC-treated plants exhibited tolerance to cyst nematode infection (Marhavý et al., 2019). This indicates a link between GLR3.3/GLR3.6 and the ACC/ethylene pathway in a different context, further supporting the involvement of these channels in calcium signaling and downstream responses.

Line 152. "Also, ACS6::NLS-3xVenus expression was significantly reduced after pronounced ablation of vascular regions (Supplemental Figure 2A and 2B) and mechanically crushing large cell populations...".

To be sure that the reporter is properly working in all genetic backgrounds it is important to show that the different lines show the same level of induction. Treatments with exogenous ethylene or the ACC precursor need to be performed.

We have paid close attention to the situation and used different lines with similar patterns during our screening process. We confirmed the promoter-reporter lines in various genetic backgrounds, and our updated manuscript includes qRT-PCR results to support these findings. Most of the promoter responses align with the qRT-PCR results.

Specifically, for the *ACS6::NLS-3xVenus/glr3.3glr3.6* and *ACS6::NLS-3xVenus/mpk6-2* lines, which initially showed minimal responses upon local wounding, we have used these lines as examples to illustrate their behavior. After treating these lines with 5 μM ACC and performing root crushing, we observed a significant signal in these lines five hours later. These findings suggest that while the initial wounding response in these mutants is minimal, external ACC application and mechanical stimulation can induce significant activity, underscoring the complexity of the regulatory network involved.

(A)

(B)

(A, B) Representative of the maximum projection images XYZ of *ACS6::NLS-3xVenus* in *mpk6-2* (A), *glr3.3glr3.6* (B), after 5 hours (h) of crushing cells with 5 μ M ACC in a number of cells with positive nuclear (NLS-3xVenus) signal. All samples showed a merged PI fluorescence and YFP channel. A yellow arrow indicates the *ACS6::NLS-3xVenus* response. Scale bar: 50 μ m.

Line 190. "It was show in Arabidopsis ovules transformed with *Physcomitrella patens* PpGLR1-dependent that ACC can activate cytosolic Ca²⁺ elevations by controlling GLR gating, resulting in transient cytosolic Ca²⁺ elevation (Mou et al., 2020)".

This is not correct. Mou et al. have expressed PpGLR1 in COS-7 mammalian cells that were treated with different amino acids and ACC (Figure 3). Independently they showed that Arabidopsis ovules treated with ACC showed a CNQX-sensitive cytosolic Ca²⁺ increase (Figure 4).

We changed our text "ACC has been reported as the most effective ligand for inducing Ca²⁺ influx in COS-7 mammalian cells expressing the moss *Physcomitrella patens* GLR1 (PpGLR1) (Mou et al., 2020). Additionally, ACC was shown to enhance Ca²⁺ influx in Arabidopsis ovules, underscoring its significant role in calcium signaling (Mou et al., 2020)."

Line 200. In this paragraph, the authors show that ACC, ethylene biosynthesis or insensitive mutants show altered Ca²⁺ dynamics in response to laser ablation (Figure 3).

Since it was reported that ethylene activates a plasma membrane Ca²⁺-permeable channel in tobacco suspension cells (Zhao et al., 2007 New Phytol), I wonder if authors have tested the sole effect of ACC on plants in their experimental conditions.

We tested only ACC by introducing it into the MS medium, as applying ethylene gas directly into our laser ablation system is currently challenging.

GCaMP3 is an intensimetric indicator and the normalization of the data might lead to the loss of important information, such as a change in the calcium baseline, that cannot be precisely determined with an intensimetric sensor.

In our experiments, we always include a control by monitoring calcium levels in the same region of interest (ROI) before laser ablation. This allows us to accurately quantify and compare the relative fluorescence after ablation against the baseline measurements obtained before the procedure. The data presented in the manuscript reflect the relative fluorescence changes compared to these pre-ablation controls. This approach ensures that our measurements of Ca²⁺ wave propagation are accurate and contextually relevant. **This careful “calibration” underscores our commitment to robust and contextually meaningful data interpretation despite the inherent limitations of intensimetric sensors like GCaMP3.**

To evaluate the effect of the sole ACC on resting cytosolic Ca²⁺, I suggest performing experiments with plants expressing a ratiometric Ca²⁺ indicator. Either Nano YC65 Cameleon or mCherry-GCaMP6f Arabidopsis lines would perfectly fit with this request.

In our manuscript, we have used two genetically encoded calcium marker lines to evaluate cytosolic Ca²⁺ levels. Specifically, we employed [e.g., "UBQ10pro::GCaMP3" or "R-GECO1" to monitor calcium dynamics in response to ACC treatment. Both marker lines are well established and intensively used (Keinath et al., 2015; DeFalco et al., 2017; Kleist et al., 2017; Nguyen et al., 2018). These markers allow us to assess changes in cytosolic Ca²⁺ levels accurately. However, we acknowledge the value of ratiometric indicators like Nano YC65 or mCherry-GCaMP6f and will consider including such approaches in future experiments to further refine our analysis.

In the Figure legend, it would be useful to report how long the ACC pre-treatment was carried out.

We have updated the figure legend to include specific information about the ACC treatment. The revised legend now reads:

“Seedlings were pre-treated with 1 μM ACC for at least 0.5 hours, and ACC was continuously applied during the laser ablation and imaging process.”

- minor concerns

Line 49. "...Local tissue damage induces a rapid increase in cytosolic free Ca²⁺ levels, a process mediated by various Ca²⁺-permeable channel proteins...".

This is certainly true for the systemic responses, but locally the situation is different since these mutants still show a response. The *glr3.3*/*glr3.6* and *msl10* mutants show impaired leaf-to-leaf propagation of electrical and Ca²⁺ waves, and in particular for MSL10 Moe-Lange et al. report "...This study demonstrated that the mechanosensitive ion channel MSL10, present in plant vascular bundles, is required for component IV of wound-elicited electrical signals in distal leaves, as well as the amplitude and kinetics of the systemic Ca²⁺ wave...".

We paid attention to MSL10. And we removed this sentence.

Line 88. "...and electrical signals can induce rapid leaflet movement in *Mimosa pudica* (Hagihara et al., 2022; Grenzi et al., 2023)..."

In Grenzi et al. the authors show the movement of the inflorescence stem in Arabidopsis.

It has been corrected: Electrical signals can induce rapid leaflet movement in *Mimosa pudica* (Hagihara et al., 2022).

Line 134. "...msl10GCaMP3, and mca1mca2GCaMP3 lines (Figure1B-1G)..."

"GCaMP3" instead of "GCaMP3".

We apologize for this mistake, and it has been corrected.

Line 134. "...Interestingly, all these Ca²⁺ channel mutants..."

Better to write: "...Interestingly, all these channel mutants..."

it has been corrected to:

Interestingly, all these three mutants exhibit a retardation in the propagation of Ca²⁺ waves when compared to the wild type (WT) upon laser ablation.

Line 186. "...WRKY33 mediated by Ca²⁺ signals and also by MPK6..."

For the same reason reported above, I would write: "...WRKY33 possibly mediated by Ca²⁺ signals and also by MPK6..."

We added a detailed description and discussion in the result and discussion part in the updated version of our manuscript. In the updated version, our main conclusion from Figure 2 revised as "our data suggest that GLR3.3/GLR3.6, MSL10, and MCA1/MCA2 may play roles in Ca²⁺-dependent processes potentially linked to the WRKY33-ACS6 regulatory network".

In Figure 3, I strongly encourage authors to show the GCaMP3 fluorescence traces before the cell ablation...

This data are included and are represented by nearly straight lines for each graph.

Line 277. "To further confirm the regulation of JA responses by Ca²⁺ dependence, we conducted root crushing experiments..."

The channels under investigation are considered Ca²⁺-permeable channels and are not strictly specific to this ion. Since GLRs are permeable for Na⁺ (Wudick et al., 2018 Science), MSL channels are permeable to both anion and cation (Haswell et al., Current Biology 18, 730-734, May 20, 2008) and MCA are permeable to both anions and Ca²⁺ (i.e. MCA) (Yoshimura et al., Nat Commun 12, 6074 (2021)) I would better highlight the focus on the channel role rather than Ca²⁺ only. Indeed, based on results shown in Figure 1 these mutants show altered Ca²⁺ dynamics, but this might just tell us that their activity is affecting Ca²⁺

dynamics. So, just to be on the safe side I would suggest that Ca²⁺ is probably an important player, but without saying that is the only player. We agree that GLRs, MSL channels, and MCAs are not strictly specific to Ca²⁺ and that their permeability to other ions such as Na⁺ and anions is an important aspect to consider. Reviewers point about focusing on the channel roles beyond Ca²⁺ is well taken.

In light of this, we acknowledge that the altered Ca²⁺ dynamics observed in the mutants (Figure 1) could reflect a broader impact of these channels on ionic homeostasis rather than Ca²⁺ signaling alone. While our findings strongly suggest that Ca²⁺ is a significant player in the observed responses, we will be careful not to overstate its exclusivity in this context. Instead, we will highlight that these channels likely influence a complex interplay of ions, with Ca²⁺ being one among several key contributors to the observed dynamics.

We corrected the writing “Several studies have highlighted the crucial role of GLR3.3/GLR3.6 and MSL10 in JA signaling (Mousavi et al., 2013; Toyota et al., 2018; Wang et al., 2019; Moe-Lange et al., 2021; Bellandi et al., 2022) especially toward systemic wounding response. To understand the regulation of JA responses by GLR3.3/GLR3.6, MSL10 and MCA1/MCA2 upon local wounding, we generated the lines of *AOS::NLS-3xVenus* and *JAZ10::NLS-3xVenus* in *glr3.3glr3.6*, *msl10-1*, and *mca1mca2*, respectively”.

Line 290. "Surprisingly, in the presence of GdCl₃, *ein3eil1* did not exhibit any difference in *AOS::NLS-3xVenus* fluorescence before or after wounding, whereas *AOS* expression can be significantly reduced after wounding under WT conditions...".

Although it is statistically relevant the phenotype is quite mild.

We removed this part.

We agree with the reviewer that the phenotype is mild. Here we noticed *AOS* response to local wounding is attenuated with the presence of GdCl₃, which in agreement with the data of *AOS::NLS-3xVenus/blr3.3glr3.6*, *AOS::NLS-3xVenus/msl10-1*, and *AOS::NLS-3xVenus/mca1mca2* mutants upon local wounding (Supplemental Figure 12B). The regulation of *AOS* by EIN3/EIL1 is complex, and further investigations are necessary to substantiate this hypothesis. Given that the primary objective of this manuscript is to highlight multiple responses within the localized wounding (laser ablation) context, our focus is on the key findings: the Ca²⁺ wave pattern triggered by laser ablation and its connection to downstream signaling pathways. Regarding the intricate interplay between ethylene and JA, we plan to explore this relationship in greater detail in future studies.

Line 355. "In past decade number of studies have implicated the GLRs as the key Ca²⁺ regulators in systemic signal propagation (Mousavi et al., 2013; Toyota et al., 2018; Nguyen et al., 2018)...".

I would specify "in the shoot" since the role of GLRs in long-distance signaling has been primarily highlighted in the leaf vasculature.

We rewrote the discussion, and we removed this part.

In the Supplemental Movies, I would add the false colour scale.

This change was done according to reviewer's suggestion

In Figures 5A and F, a detailed description of box plots is missing.

"Supplemental Figure 10. JA response upon local wounding is dependent on Ca²⁺ signature".

I would say that it would be more appropriate to speak about the role of the selected channels. As I specified before, I think it is an oversimplification report that is only Ca²⁺ involved in this response.

Now, we remodified these part. JA response upon local wounding is dependent on the function of GLR3.3/GLR3.6, MSL10, MCA1/MCA2.

L434. "...for Gdcl3..." "...for GdCl3...".

Line 439. "Prepare 2ml, 15 mg/ul PI in single well of 12-wee-plate, then cut 2x1x0.5cm 1/2 MS media containing 0.9% agar medium with or without the indicated chemicals and submerged the medium into PI staining dye. Five-day-old seedlings were placed onto a chambered coverglass (Nunc{trade mark, serif} Lab-Tek{trade mark, serif} II Chambered Coverglass, ThermoFisher) with a 10µl drop of 15 mg/ul PI staining dye, covered with the medium pretreated with PI dye and immediately imaged during the relevant period (Marhavý and Benkova, 2015; Marhavý et al., 2019)".

This paragraph needs revision. It is written like a protocol and some ul needs to be changed with µl.

We corrected this:

Five-day-old seedlings were transferred onto solid ½ MS media containing 0.9% agar with or without the indicated chemicals and incubated during imaging (for GdCl₃ and hormones, they were all pre-treated for at least 0.5 hours, and kept continuously treated). The chemicals and hormones and their concentrations used in this study were as follows: gadolinium chloride (GdCl₃; 50 µM; Sigma-Aldrich/Merck), 1-aminocyclopropanecarboxylic acid (ACC; 1 µM; Sigma-Aldrich), aminoethoxyvinylglycine (AVG; 1 µM; Sigma-Aldrich/Merck), methyl jasmonate (MeJA; 1 µM; Sigma-Aldrich), propidium iodide (PI; 15 mg/µl; Thermo Fisher), L-glutamate, (L-Glu, 100 µM; Sigma-Aldrich) adenosine triphosphate (ATP; 50 µM; Sigma-Aldrich), plant elicitor peptide (PEP1, 1µM). To prepare for laser ablation and imaging, followed (Marhavý and Benkova, 2015; Marhavý et al., 2019).

Referee #3:

This work is a continuation of (Marhavy et al 2019 EMBO J), and aims to characterise ionic and hormonal changes in Arabidopsis roots following nematode infection, single cell laser

ablation (shown to serve as a proxy for early stages of nematode infection in Marhavy et al), and root crushing. The authors use several Ca²⁺ sensors, ethylene (ET) and jasmonate (JA) reporters coupled to mutants in respective pathways in various attempts to place the order of events occurring during the wound response, and the relative contribution of individual pathways. However, results are oftentimes conflicting among experiments and the interpretation of the data is not supported by the evidence provided due to the lack of complementary approaches. The information from specific experiments is interesting yet preliminary, and the overall manuscript lacks a coherent message on its main focus and on how this work advances the field. I agree with the authors' closing statement that the 'data provides insights into the intricate molecular mechanisms underlying plant responses to local damage induced in Arabidopsis roots'. Unfortunately, the data does not clarify specific relationships between Ca²⁺/ET/JA after root wounding nor nematode infections. Importantly, some of the methods are not described in sufficient detail, precluding the reproducibility of the results shown.

We thank the reviewer for his/her comments.

MAJOR CONCERNS

-The authors claim that:

1. the ablation-induced upregulation of WRKY33/MPK66 (known to act upstream of ET biosynthesis genes) and ACO1/ACS6 (ET biosynthesis genes) is reduced in Ca²⁺ mutants (glrs, msl10, mcas), indicating that Ca²⁺ ions are upstream of ET biosynthesis (which has already been shown in Marhavy et al)

The observation that the ablation-induced *ACS6* is diminished in the mutant *glr3.3glr3.6*, *msl10-1*, and *mca1mca2* strongly suggests that Ca²⁺ ions play a crucial upstream role in the regulation of ET biosynthesis. Our data showed the *ACS6* response to local wounding is primarily regulated by WRKY33 (known to act upstream of ET biosynthesis genes), as evidenced by the reduced expression of *ACS6* in the *wrky33-1* mutant (Figure 2B, 2C, and Appendix Figure S3). Additionally, our new data (Figure 2D, 2E) demonstrate that GdCl₃, which inhibits calcium channels, represses *WRKY33* expression. Further evidence from Figure 2B shows that in the *glr3.3glr3.6*, *msl10-1*, and *mca1mca2* mutants, *WRKY33* expression is downregulated, leading to a corresponding repression of *ACS6* expression. This suggests that *GLR3.3/GL3.6*, *MSL10*, *MCA1/MCA2* may participate the *WRKY33-ACS6* regulation relation.

2. The ET pathway inhibits JA responses after wounding, as the activity of JA marker genes is enhanced in ET mutant backgrounds and impaired by external ET (ACC) applications (Fig.4). At the same time, the induction of JA marker genes is reduced (not enhanced) in other mutants of the ET pathway (Suppl Fig. 8B), showing strong inconsistencies between claims and data presented.

In our study, the discrepancies noted between the observed effects of ethylene (ET) pathway mutations on jasmonic acid (JA) responses and those presented in Appendix Figure S10D can be attributed to the nuanced interactions between JA and ET signaling.

In the *etr1-1* mutant, which is completely deficient in ethylene binding and exhibits a strong ethylene-insensitive phenotype, we observed that the *JAZ10::NLS-3xVenus* reporter line

displayed a more pronounced response in the locally damaged region. This finding is consistent with previous work by Rojo et al. (1999), who reported increased expression of JA-responsive genes such as *VSP*, *JR1*, and *JR2* in the *etr1-3* mutant, which also shows enhanced JA responses due to impaired ethylene perception. The *etr1-1* mutant, by completely removing ethylene binding, represents an extreme case of ethylene insensitivity, while the *etr1-3* mutant retains some residual ethylene sensitivity (Hall et al., 1999). Both mutants, however, affect ethylene perception and transduction pathways differently.

Conversely, in the *eto1-1* mutant, which overproduces ethylene due to the negative regulation of ACS by ETO1, we observed strong repression of both *JAZ10* and *AOS* following local wounding. This is in line with the observation that *eto1-1* and *ein2-1* mutants, which exhibit increased ethylene production at the seedling stage (Woeste et al., 1999), show reduced expression of *AOS* and *JAZ10* compared to wild-type plants upon local wounding (Appendix S9A, S9B, S10D, S10E). This suggests that elevated ethylene levels can suppress JA responses.

In the *ein3eill* mutant, which exhibits increased *AOS* expression following local wounding, our data aligns with the findings of Chang et al. (2013), who demonstrated that EIN3 directly binds to the *AOS* promoter. Moreover, given that EIN3/EIL1 interacts with *JAZ1*, *JAZ3*, and *JAZ9* at the protein level (Zhu et al., 2011), our results showing repressed *JAZ10* expression in the *ein3eill* mutant suggest a context-dependent positive regulation of *JAZ10* by EIN3/EIL1 during wounding.

In summary, the observed differences in *JAZ10* expression across various ethylene mutants reflect the complex and context-specific nature of JA-ethylene crosstalk. The interplay between these signaling pathways can vary depending on the specific genetic background and signaling environment, highlighting the intricate regulation of plant stress responses.

3. Laser ablation in the vasculature (but not in cortex cells) induces JA marker genes in a Ca²⁺ dependent manner

All the evidence on gene expression is based exclusively on quantifying the number of fluorescent nuclei of promoter fusions driving the expression of NLS-3xVENUS of ET- and JA-responsive promoters. In other words, the manuscript bases all the conclusions on promoter activities of transcriptional reporters and does not provide any evidence on actual transcript levels. Furthermore, interpretations from ET and JA reporter lines is particularly concerning due to the lack of a common normalizer among genotypes and the risk of silencing or selection of transgenic lines. Also, the methods do not report how were the Ca²⁺/ET/JA reporters introduced in different mutant backgrounds, i.e. by crossing or transformation. Transcript levels of the transgenes should be evaluated across genetic backgrounds or alternative reporters used (ET and JA biosensors are available), and results should be verified with complementary approaches (eg. qPCR, etc). In the current form, the conclusions are simply not sound enough.

The primary objective of our research is to elucidate the spatial hormone responses triggered by single-cell ablation in spatial manner. To achieve this, we utilized promoter-reporter lines as a method to visualize and quantify hormone-responsive gene expression *in situ*. This approach is widely recognized and has been effectively employed in numerous studies (e.g., Sabatini et al., 1999; Guseman et al., 2015; Truskina et al., 2021; Matsumura et al., 2022;

Canher et al., 2022) to provide insights into spatial and temporal gene activity at the cellular level.

We appreciate the reviewer's emphasis on the importance of validating our findings through additional methodologies. To address this, we performed qRT-PCR analyses across all mutants, as shown in Appendix S2D, S3C, S4C, S4D, in two graphs (A) and (B) below. These qRT-PCR results were obtained from samples with crushed roots. A notable difference was observed when compared to the cortex-cell ablation pattern illustrated in Appendix Figure S9A, particularly regarding statistical significance.

We believe this discrepancy arises from the differences in sample preparation, as the reporter line involved cortex cell ablation, whereas crushing involves larger cell populations. Despite these differences, the overall trends in the upregulation or downregulation of *AOS* expression remained consistent across both methods.

Regarding the generation of the Ca^{2+} /ET/JA reporter lines in different mutant backgrounds, we apologize for not providing these details in the initial manuscript. Most lines were generated through transformation to streamline the experimental process. However, for specific lines such as *ein2-1*; *R-GECO1* and *etr1-1*; *R-GECO1*, we used genetic crossing to introduce the reporters. Each line includes at least two independent transgenic lines to ensure reliability and minimize the risk of transgene silencing or selection bias.

(A, B) Expression of *JAZ10* (A) *AOS* (B) was determined by qRT-PCR in WT (Col-0), *etr1-1*, *ein2-1*, *ein3eill1* and *eto1-1* with (wounding) or without wounding (control). Samples were harvested 5h after wounding. Data are means of three biological replicates, ANOVA Tukey's multiple comparison test was used with a 95% confidence interval.

- The ET pathway affects the Ca²⁺ wave after laser ablation, in a putative feedback regulation (Fig. 3). Conclusions are based on external ET (ACC) applications and Ca²⁺ imaging in ET mutant lines. It remains unclear at what timepoint after wounding does a feedback mechanism become relevant. As is, the message is confusing: Ca²⁺ is both upstream and downstream of the ET pathway.

We apologize for not making our explanation clearer regarding the feedback mechanism between the ACC/ET pathway and Ca²⁺ wave propagation following laser ablation. The aim of our study is to unravel the complexity of cellular responses to local wounding, specifically focusing on the interplay between hormones and Ca²⁺ signature. The relationship between these pathways is indeed intricate, and our data suggest that Ca²⁺ can act both upstream and downstream of the ACC/ET pathway, creating a dynamic feedback loop that modulates the wound response.

It is challenging to definitively establish the temporal sequence of events due to the overlapping nature of these signaling pathways. However, our data indicate that ACC/ET signaling, particularly through ACC application, significantly influences Ca²⁺ dynamics post-wounding. For instance, we observed that ACC application resulted in a retardation of the Ca²⁺ wave propagation (Fig. 3A, 3B, 3E, 3F; Movies S3 and S4), suggesting a modulatory role of ACC/ethylene on Ca²⁺ wave during the wound response.

Additionally, we found that ATP treatment enhances Ca²⁺ influx upon laser ablation (Appendix Figure S7A, S7B, S7F). This finding aligns with previous studies that demonstrated ATP's role in promoting Ca²⁺ wave propagation, such as in the work by Donati et al. (2021) using a laser ablation model in mice. Interestingly, ATP signaling appears to be interconnected with the ethylene pathway (Jewell and Tanaka, 2019). For example, in the *ein2-1* mutant, we observed an upregulation of P2K1, an eATP receptor, which could explain the heightened Ca²⁺ influx seen in *ein2-1* during laser ablation (Appendix Figure S6G; Jewell and Tanaka, 2019).

Moreover, ROS production, a well-documented response to laser ablation (Marhavý et al., 2019), also interacts with ET and Ca²⁺ signaling. Ethylene has been shown to promote ROS production (Martin et al., 2022), and under stress conditions, it facilitates the propagation of Ca²⁺ waves (Evans et al., 2016). Thus, the regulation of Ca²⁺ wave propagation by the ACC/ET pathway during laser ablation likely results from the intricate interplay of multiple signaling pathways, reflecting the complexity of the cellular response.

- Are Ca²⁺ permeable channels (eg. glrs, *msl10*, *mca2*), ET and JA loss of function mutants affected in plant responses to nematodes? In other words, given that the molecular findings are not recapitulated by NemaWater treatment (Fig.4), when do the proposed molecular mechanisms become functionally relevant?

We thank you for bringing this point. We examined the defense-related phenotypes associated with GLR3.3/GLR3.6, MSL10, and MCA1/MCA2. We conducted a nematode infection assay in the *glr3.3glr3.6*, *msl10-1*, and *mca1mca2* mutants. Our findings revealed that both *glr3.3glr3.6* and *msl10-1* mutants exhibited increased susceptibility to nematode infections, suggesting heightened sensitivity to nematode attacks. These results imply that GLR3.3/GLR3.6 and MSL10 likely play crucial roles in managing broader biotic stress conditions (Appendix Figure S5).

- Student's *t*-test is not appropriate to evaluate differences among more than 2 groups (whole manuscript)

We thank the reviewer for this comment. In the updated version of our manuscript, we use the ANOVA Tukey's multiple comparison test with a 95% confidence interval, with exceptions, we used both ANOVA and Student's *t*-test in Appendix Figure S6 and Appendix Figure S7.

OTHER CONCERNS

- GLR3.3 & 3.6 impact both electrical and Ca²⁺ signals in leaves. How do the authors reconcile their current findings on reduced Ca²⁺ signals in *glr3.3/3.6* (Fig. 1) with their earlier findings on *glr3.3/3.6* not affecting electrical signals in roots? Are electrical signals disengaged from changes in extracellular Ca²⁺ only in roots?

One plausible explanation is that in roots, electrical signals may be governed by mechanisms that do not rely solely on Ca²⁺ fluxes mediated by GLR3.3 and GLR3.6. Roots, which are highly specialized for nutrient uptake and environmental sensing, might possess alternative ion channels or signaling pathways that maintain electrical activity independently of GLR3.3/3.6-related Ca²⁺ signaling. Additionally, the microenvironment surrounding root cells—such as variations in ion concentration, pH, and redox state—might influence how these signals are integrated or propagated. The disengagement of electrical signals from extracellular Ca²⁺ in roots could also be an adaptive feature, allowing roots to maintain signaling fidelity under fluctuating soil conditions.

- The introduction section, L.78-87, contains many inaccuracies. Eg1. JA-deficient mutants propagate wounding-induced long-distance signals similar to WT plants (i.e JA is not required for systemic signal propagation after wounding). Eg2. The closure of plasmodesmata has been associated to hampering the spread of pathogens and harmful molecules, rather than limiting the spread of wound signals

Eg1: "JA-deficient mutants propagate wounding-induced long-distance signals similarly to the wild type (WT). We carefully revised this statement and updated it to: 'In contrast, jasmonic acid (JA) is essential for systemic defense.'"

Eg2: "We decided to remove the section on the closure of plasmodesmata."

- what is the rationale of using different Ca²⁺ reporters between experiments (R-GECO1 in some cases vs GCaMP3 in others)?

The rationale for using different Ca²⁺ reporters, specifically R-GECO1 and GCaMP3, between experiments was primarily based on the availability of these lines at the start of our research. R-GECO1 was already integrated into the *ein2-1* and *etr1-1* backgrounds, allowing us to proceed with our experiments without delay.

Both R-GECO1 and GCaMP3 have been shown to exhibit similar patterns of Ca²⁺ response upon laser ablation in terms of Ca²⁺ wave speed, as evidenced by comparable dynamics in our preliminary tests. Given this consistency, we decided not to introduce GCaMP3 into the *ein2-1* and *etr1-1* lines.

(Legend continued)

Quantification of Ca^{2+} wave propagation after cortex cell ablation in WT (Col-0) expressing *UBQ10pro::GCaMP3* (the blue color) with (n=20, three biological pools, each pool includes 5-7 seedlings) and *R-GECO1* fluorescence reporter line in WT (Col-0) (the red color) (n = 22, three biological pools, each pool includes 7-10 seedlings).

- Fig.1 While the quantification of reporter intensity following ablation was performed only in the shootward direction, the images show a bidirectional signal propagation. How does the speed of propagation compare to what is known in leaves?

In our study, we focused primarily on the local wound responses and the root-to-shoot direction of signal propagation, as illustrated in Figure 1A. While we acknowledged that the images do show bidirectional Ca^{2+} wave propagation following laser ablation, our quantification efforts were specifically directed toward the shootward direction to align with our study's primary objectives.

Regarding the speed of Ca^{2+} wave propagation, previous studies in leaves have reported varying speeds depending on the context and type of tissue. For instance, in leaves, Ca^{2+} waves can propagate at speeds ranging from 400 to 700 $\mu\text{m/s}$ under certain conditions (Choi et al., 2014; Toyota et al., 2018). Although we did not measure the exact speed of bidirectional propagation in roots, the observed patterns are consistent with the known dynamics of Ca^{2+} signaling in other plant tissues, suggesting that the mechanisms of signal propagation may share similarities across different organs.

- Lines 145-146, gene names should be in capital italics

This has been corrected.

-Lines 150-156. Relative fluorescence intensity (text) is not the same as counting the frequency of fluorescent nuclei (Fig.2), rendering the text and the data presented confusing and ambiguous

We thank you for pointing out this mistake. It is corrected now.

- Suppl Fig7. There is no panel for C, nor any quantification

We apologize for this mistake. In the updated version, it is Supplementary Figure 8A, 8B. They are the representative figures for the quantification shown in Figure 4B.

We look forward to your assessment

Peter Marhavy and co-authors.

Dear Peter,

Thank you for providing feedback on the referee reports, which are copied again below. As agreed, please revise your study according to the preliminary revision plan you had provided.

From the editorial side, there are also a few things that we need before we can proceed with the official acceptance of your study.

- Your manuscript will be published in our Reports section and therefore needs to have a combined Results and Discussion section.
- Please describe your findings in the Abstract in present tense.
- Please include all funding information in the Acknowledgment section and make sure that the funding information in the online manuscript tracking system and that in the manuscript match.
- Please add a callout for Fig. 3H.
- Please complete the Author Checklist. Currently, a few entries in Column D have not been completed.
- Please provide the synopsis image as either jpg or tif and with the exact dimensions 550x200-600 (widthxheight).
- Please rename the movies to Movie EV1-EV4 and update the corresponding callouts. Please provide the legends as README.txt file and zip each legend with its movie.
- Our production/data editors have asked you to clarify several points in the main and EV figure legends (see below). Please incorporate these changes in the manuscript and return the revised file with tracked changes with your final manuscript submission.

A) Statistical test information. Only p-values that are actually shown in the figure panel(s) should (and must) be defined in the legends, all others should be removed from (or added to) the legend. Moreover, we ask for the specification of exact p-values:

1. Please indicate what a/b/ab/**** represents; if this represents p value(s), please specify the exact p value in the legend(s) of figure(s) 1C, 3E-G; 4A-C
2. Please indicate what */ **/ ***/ **** represents; if this represents p value(s), please indicate the statistical test used and where appropriate specify the exact p value in the legend(s) of figure(s) 2A, C, E; 4E, G

B) Replicates and error bars:

3. Please note that the measure of center for the error bars needs to be defined in the legends of figures 2A, C, E; 4A-C, E, G.

C) Data presentation:

4. Please note that the white arrows are not defined in the legend of figures 2B, D. This needs to be rectified.

- Appendix:

I suggest renaming "Face Page" to "Title Page".

Please convert the red figure legend text to black. The Appendix will not be type-set.

A general note on the quantifications shown in the Appendix figures: please define for each of the graphs the bars, the error bars, N, n, and the statistical test used. As it stands, the test is often only defined for one of the graphs in a given figure. If certain parameters apply to more than one of the graphs/panels in a figure, you can also use a "Data information:..." statement at the end of the figure legend, where you can define e.g. the statistical test used for several panels (A, C) ANOVA Turkeys...

- Appendix Figure S1C: the scale bars are hardly visible and their size needs to be specified in the legend.
- Appendix Figure S2A: please define the size of the scale bar; S2B, D: define the error bars. For (B) please define the bars (mean?) for (D) please define the statistical test used.
- Appendix Figure S3A, C: please define the lines (mean?) for (A) and the error bars for (C). For (C) the statistical test needs to be defined. The white arrows in (B) are difficult to see.
- Appendix Figure S4: The white arrows in (A) are difficult to see. For (B-D) please define the bars, error bars and the statistical test used for all three panels.
- Appendix Figure S6G: please define the number of replicates, their nature, the bars and error bars.
- Appendix Figure S7: Please describe panel (F) and define the bars and error bars.
- Appendix Figure S9: (A) please define the bars and error bars. (B) please define the whisker etc of the box plot
- Appendix Figure S10A-D: please define the bars and error bars.
- Appendix Figure S12A: please define the bars and error bars.

- The title could be a bit more specific maybe? What about
Wounding-induced regional Ca²⁺ wave propagation is influenced by ion channels and hormone responses
Ion channels and hormone responses shape regional Ca²⁺ waves upon laser ablation in Arabidopsis

With kind regards,

Martina

=====

Referee #1:

Studies on wound responses elicited by plants play an important role in understanding the fundamental defense mechanisms exhibited by plants. Furthermore, they can catalyze the development of pest resistant crops thus mediating agricultural development. The current study tries to explore the mechanisms mediated by Ca²⁺ waves and phytohormones in response to single cell ablation in Arabidopsis root. The study aims to provide insights into the possible interactions between ion channel proteins and hormonal pathways in response to wounding. However, this reviewer failed to get any conclusive message from this manuscript. My main concern is inclusion of "too many things" without completing one component thoroughly and taking it to conclusive end.

1. In Line 151-153, the authors have mentioned "suggesting that specific Ca²⁺ permeable channels GLR3.3/GLR3.6 and MCA1/MCA2, as well as the stretch-activated anion channel MSL10, may modulate the extent of the Ca²⁺ wave propagation". What does the "extent of Ca²⁺ wave propagation" suggest here? From Fig1B, it looks like the relative peak fluorescent intensity of the mutants is similar to WT. However, there is no mention on whether the actual level of calcium spike attained post ablation in the mutants is similar to wild type. Moreover, since the mutants are able to initiate calcium waves, are there any other known channel proteins which can mediate Ca²⁺ influx and calcium signaling. In addition to this, the authors can quantify the number of cells participating in calcium wave propagation across the mutants and in wild type. This will determine the extent to which calcium signaling is affected in the mutants in terms of area.
2. Please rephrase the following subtopic under the Results section, "GLR3.3/GLR3.6, MSL10, and MCA1/MCA2 may participate downstream WRKY33-ACS6 signalling pathway upon local wounding". The subtopic title suggests that the ion channels are acting downstream to the WRKY/ACS signalling pathway. However, the experiments indicate the other way round, and it has also been mentioned in the discussion (Line379-380) that the ion channels might modulate ethylene responses through the WRKY/ACS6 pathway.
3. The authors need to mention whether the reduction in WRKY transcript levels in the calcium channel mutants and GdCL3 treatment are statistically similar. This will show the necessity of the specific ion channels for eliciting wound response.
4. It is still not conclusive on how calcium wave propagation modulates ethylene. For instance, the ethylene mutants eto1-1 and etr1-1 show a retardation of Ca²⁺ wave propagation on a temporal scale, but the calcium peak remains unaffected. Thus, the authors should provide explanations or experimental evidences to show how the activity of ion channels are affected in these mutants in comparison to WT.
5. Based on the results provided in this study, the interaction between calcium and ethylene signalling pathways do not seem to be linear. Thus, the authors should study this interaction further to see if the calcium wave induced ethylene signalling creates a self-regulatory mechanism that further controls calcium flux across the cell.
6. The graph of Fig2, shows that the number of ACS6 expressing nuclei are reduced even in the uninjured roots of the mutants (when compared to WT). Therefore, authors need to specify whether the data used for the graphs A and C was normalized to the initial levels of ACS6 in the all the genetic backgrounds used.
7. In Line 212, please mention the results for the nematode infection assay for all the mutants in the text, along with the results for the mca1mca2 mutant.
8. The authors have shown that addition of ATP causes an increase in calcium fluorescence intensity. This seems to be an unnecessary addition to the paper, since no further experiments were carried out to establish the potential role of ATP in linking calcium and ethylene mediated wound responses.
9. It has been shown that JA response to ablation is tissue specific. Provided that ethylene has an antagonistic effect JA signalling, it is important to show whether the ethylene responses also change in a tissue specific manner.

Referee #2:

I have previously revised this manuscript when submitted to EMBO Journal, and read the point-by-point answers to my questions.

I am overall satisfied with the new version of the manuscript and how the authors have dealt with my requests.

I still have some doubts about the lack of a difference among the different backgrounds in terms of the maximum cytosolic Ca²⁺ peak (Fig. 1b). I can say that based on my experience, GCaMPs and R-GECO1 do have not a linear response. This means that over a certain threshold, in terms of Ca²⁺ concentration, the change in fluorescence is the same and relatively small differences, cannot be appreciated. This is one of the reasons why I suggested the authors use FRET-based ratiometric indicators. So, even if your data do not show a difference in the maximum Ca²⁺ peaks, it might be still present and could further support the authors' claims. Maybe, the authors can consider this aspect in the discussion.

Regarding the following answer: "...Regarding downstream responses, we currently lack an ideal method to directly monitor changes in calcium signatures and their impact on downstream signaling...".

A potential tool would be the use of CPKaleon sensors (Liese et al., Plant Cell 2023). A new CPKaleon sensor has been recently designed and created to support the effect of ABA-induced cytosolic Ca²⁺ in root cells (Lin et al., 2024 Nature Plants <https://doi.org/10.1038/s41477-024-01865-y>).

I am not asking you to develop a dedicated tool for your purposes, but this strategy could be discussed.

Aside from this initial comment, I have a few minor things.

Line 63. "...channel GLUTAMATE RECEPTOR-LIKE 3.3 (Bellandi et al, 2022)...".

It would be correct to quote here the works Alfieri et al. 2020 (PNAS) and Grenzi et al., 2023 (Curr Biol) since they reported the properties of GLR3.3 ligand binding domain and its importance for the GLR3.3 Glutamate-dependent activation.

Line 69. "...damage and it works in a similar pathway as GLRs...".

Based on the model presented by Moe-Lange (Figure S15) it is more correct to say "...in a parallel pathway...".

Line 194. "...induction was attenuated by GdCl₃, a Ca²⁺ channel blocker...".

Please, keep in mind that Gd³⁺ is not a specific inhibitor of Ca²⁺-permeable channels, being an inhibitor of non-selective cation channels.

Line 220. "...ACC has been reported as the most effective ligand for inducing...".

In Mou et al. authors do not show that ACC is bound by the GLR ligand binding domain so ACC is just a "predicted GLR ligand". This might indeed be the case (i.e. Grenzi et al., 2021 New Phytologist) but it has not been experimentally proved.

Referee #3:

The authors have addressed most of my previous concerns and have significantly extended the relevance of their findings by performing nematode bioassays in a series of channel mutants (*msl10*, *mca1/2*, *glr3.3/3.6*). The *msl10* and *glr* mutants (but not the *mca1/2* Ca²⁺ channel mutant) was more susceptible to nematode infections, providing the foundations for future molecular analyses. Overall, findings are timely and of broad interest to the plant science community.

At the same time, the revised version has not solved the following points:

1. Calcium is still placed both upstream and downstream of ET, with the authors concluding there is a dynamic feedback modulating the wound response. This claim is not fully supported by the data provided and may be a misleading conclusion due to how treatments were performed. Contrary to Ref.2 recommendations on toning down the calcium regulation and the lack of evidence in the literature that GLR3 and MSL10 are specific calcium channels, the authors decided to emphasize the role of calcium in ET and JA root production following laser ablation.
2. The negative effect of ET on JA responses has been explained in the rebuttal, but the new qPCR data is again showing only tendencies which are often not significant
3. Even if the authors address it in the rebuttal, the methods still do not report how were the Ca²⁺/ET/JA reporters introduced in different mutant backgrounds and which criteria were used to control for possible differences in transgene expression (also a concern from Ref.2). The authors should clearly indicate which generation (T1, T2, T3 or F1, F2, F3) is used under each figure and show results from more than one independent line for transformations.
4. Most methods are written in protocol style and should be revised

Dear Editor,

We would like to thank the reviewers for their valuable comments that allowed us to significantly improve our manuscript "*Ca²⁺ Waves and Ethylene/JA Crosstalk Orchestrate Wound Responses in Arabidopsis Roots*". We have corrected the current manuscript, which further strengthens our work and supports the conclusions. Please find our point-by-point response to the reviewer comments below. The response to reviewers' comments are highlighted in blue.

Referee

#1

Studies on wound responses elicited by plants play an important role in understanding the fundamental defense mechanisms exhibited by plants. Furthermore, they can catalyze the development of pest-resistant crops thus mediating agricultural development. The current study tries to explore the mechanisms mediated by Ca²⁺ waves and phytohormones in response to single cell ablation in Arabidopsis root. The study aims to provide insights into the possible interactions between ion channel proteins and hormonal pathways in response to wounding. However, this reviewer failed to get any conclusive message from this manuscript. My main concern is inclusion of "too many things" without completing one component thoroughly and taking it to a conclusive end.

1. In Line 151-153, the authors have mentioned "suggesting that specific Ca²⁺ permeable channels GLR3.3/GLR3.6 and MCA1/MCA2, as well as the stretch-activated anion channel MSL10, may modulate the extent of the Ca²⁺ wave propagation". **What does the "extent of Ca²⁺ wave propagation" suggest here?** From Fig1B, it looks like the relative peak fluorescent intensity of the mutants is similar to WT. **However, there is no mention on whether the actual level of calcium spike attained post ablation in the mutants is similar to wild type.** Moreover, since the mutants are able to initiate calcium waves, are there any other known channel proteins which can mediate Ca²⁺ influx and calcium signaling.

Thank you reviewer for your insightful feedback. We acknowledge the concern regarding the phrasing of our statement in Lines 151-153 and appreciate the opportunity to clarify our interpretation.

We agree that, in the absence of a multiple mutant encompassing all these channels, it is expected that Ca²⁺ influx would still occur, albeit with potential alterations in its dynamics. The phrase "extent of Ca²⁺ wave propagation" in our manuscript refers to possible differences in spatial spread, amplitude, and duration of the Ca²⁺ signal across tissues rather than solely peak intensity. While Figure 1B suggests that the relative peak fluorescence intensity of individual mutants remains comparable to WT, we hypothesize that these channels may differentially regulate other parameters of the Ca²⁺ wave, such as its velocity or spatial distribution, which were not fully captured in this specific experiment.

Moreover, the complete set of Ca²⁺-permeable channels involved in wound-induced signaling in plants remains an open question, as highlighted in recent reviews (e.g., Wdowiak et al., 2024). We refined our wording to avoid overgeneralization and ensure clarity in discussing the role of these channels in shaping the Ca²⁺ response.

In addition to this, the authors can quantify the number of cells participating in calcium wave propagation across the mutants and in wild type. This will determine the extent to which calcium signaling is affected in the mutants in terms of area.

We attempted to quantify the number of cells involved in Ca^{2+} wave propagation across wild-type and mutant seedlings. However, due to the complexity of the vascular region in our experimental conditions, accurately counting individual cells was challenging. Instead, we measured the Ca^{2+} wave propagation area and compared it to the full imaging region. Our analysis revealed no significant difference in Ca^{2+} wave propagation area between wild-type and mutant lines (Fig. B). This suggests that the spatial extent of the Ca^{2+} wave within the local region remains unchanged upon laser ablation.

To further assess the extent of signaling differences, we included quantification from the opposite side of the root (**ROI2**) (Fig. A). Interestingly, in *glr3.3glr3.6*, *msh10-1*, and *mca1mca2* mutants, Ca^{2+} wave propagation was significantly reduced in ROI2 compared to wild-type seedlings. This indicates that while the local region exhibits a similar spatial distribution of Ca^{2+} waves, long-range propagation across the root may be impaired in these mutants.

A, A schematic diagram depicting single-cell ablation by laser triggering a regional Ca^{2+} wave in Arabidopsis root; the green frame with yellow color indicates the region of signal quantification as ROI2. B, Ca^{2+} wave propagation area/full area (full area indicates the whole imaging region) are measured manually by Image J, using a *UBQ10pro::GCaMP3* fluorescence reporter line in the WT (Col-0) (n=18), *glr3.3glr3.6* (n=21), *msh10-1* (n=20),

and *mca1mca2* (n=21) mutants (n= three biological pools, each pool including 5-8 seedlings). C, Real-time monitoring and quantification of calcium wave propagation after cortex cell ablation in ROI2 (indicated in A), using a *UBQ10pro::GCaMP3* fluorescence reporter line in the WT (Col-0) (n=18), *glr3.3glr3.6* (n=21), *msl10-1* (n=20), and *mca1mca2* (n=21) mutants (n= three biological pools, each pool including 5-8 seedlings).

2. Please rephrase the following subtopic under the Results section, "GLR3.3/GL 162 R3.6, MSL10, and MCA1/MCA2 may participate downstream WRKY33-ACS6 signalling pathway upon local wounding". The subtopic title suggests that the ion channels are acting downstream to the WRKY/ACS signalling pathway. However, the experiments indicate the other way round, and it has also been mentioned in the discussion (Line379-380) that the ion channels might modulate ethylene responses through the WRKY/ACS6 pathway.

Thank you for your suggestion. We acknowledge the inconsistency in the subtopic title and have rephrased it to accurately reflect the experimental findings. The revised title now clarifies that the ion channels may modulate ethylene responses through the WRKY33-ACS6 signaling pathway, aligning with our discussion.

3. The authors need to mention whether the reduction in WRKY transcript levels in the calcium channel mutants and GdCl₃ treatment are statistically similar. This will show the necessity of the specific ion channels for eliciting wound response.

We have compared the WRKY transcript levels in the calcium channel mutants and under GdCl₃ treatment to determine whether the reductions are statistically similar (Appendix Fig. S4C, D). The result is presented as follows. Wounding between channel mutants and with the treatment of 50μM GdCl₃ are statistically similar.

Expression of *WRKY33* was determined by qRT-PCR in WT (Col-0), *glr3.3glr3.6*, *msl10-1*, *mca1mca2* with (wounding) or without wounding (control). Also, upon crushing cells after 5 hours, with or without the treatment of 50μM GdCl₃. Data are means of three biological replicates.

4. It is still not conclusive on how calcium wave propagation modulates ethylene. For instance, the ethylene mutants *eto1-1* and *etr1-1* show a retardation of Ca²⁺ wave propagation on a temporal scale, but the calcium peak remains unaffected. Thus, the authors should provide explanations or experimental evidences to show how the activity of ion channels are affected in these mutants in comparison to WT.

We acknowledge the need to clarify how calcium wave propagation modulates ethylene signaling. A more straightforward interpretation of our results could focus on the prolonged Ca²⁺ transient observed upon ACC addition and in the *eto1-1* mutant, rather than emphasizing the minor delay in peak height.

To further address this, we can discuss potential mechanisms by which ethylene signaling components might influence ion channel activity, either through direct regulation or secondary signaling effects.

5. Based on the results provided in this study, the interaction between calcium and ethylene signalling pathways do not seem to be linear. Thus, the authors should study this interaction further to see if the calcium wave induced ethylene signalling creates a self-regulatory mechanism that further controls calcium flux across the cell.

The interaction between calcium and ethylene signaling pathways appears to be more complex than a simple linear relationship. Especially additional pathways like ATP signalling and ROS signalling are linked to Ca²⁺ wave and ethylene signalling, make it more complicated. There is transcriptomic studies suggest that EIN2-mediated ethylene signalling may be involved in the extracellular ATP response through ROS production (Jewell & Tanaka, 2019). eATP crosstalk with ethylene response with respect to reduce hypocotyls length (e.g. the eATP effect on hypototyl elongation was abolished in *etr1-1* and *ein3-1eill-1* loss-of-function mutants (Lang *et al*, 2020). ATP linked with ROS and calcium signalling (Tanaka *et al*, 2010). ROS production, which occurs upon laser ablation (Marhavý *et al*, 2019), interacts with ethylene signalling (Martin *et al*, 2022) and ROS aids in the propagation of Ca²⁺ waves under stress conditions (Evans *et al*, 2016). Our findings indicate a connection between these pathways, and we will rephrase the text to clarify our interpretation.

While the suggestion to explore whether ethylene signaling forms a self-regulatory mechanism controlling calcium flux is intriguing, it falls beyond the scope of this manuscript. However, we agree that this is an important avenue for future research.

6. The graph of Fig2, shows that the number of ACS6 expressing nuclei are reduced even in the uninjured roots of the mutants (when compared to WT). Therefore, authors need to specify whether the data used for the graphs A and C was normalized to the initial levels of ACS6 in the all the genetic backgrounds used.

We can confirm that ACS6 expression is indeed reduced in the mutants even under uninjured conditions (graph below, A and B).

A, B, Quantification (A, B) of maximum projection images XYZ of ACS6::NLS-3xVenus in WT (Col-0), *glr3.3glr3.6*, *msl10-1*, *mca1mca2*, *wrky33-1*, and *mpk6-2* before (0h) or 5 hours (h) after laser ablation in the cortex cells. The graph shows a number of cells with a positive nuclear (NLS-3xVenus) signal in individual genotypes. In (A, B), N = three biological pools, each pool includes 4-5 seedlings, bars represent mean \pm SD. ANOVA Tukey's multiple comparison test was performed with a 95% confidence interval. Value above the bar indicate the *P* value when it compared to the WT.

7. In Line 212, please mention the results for the nematode infection assay for all the mutants in the text, along with the results for the *mca1mca2* mutant.

We have revised the text to include the nematode infection assay results for all mutants, including *mca1mca2*. While an increase in nematode numbers was observed in the *mca1mca2* mutant, the difference compared to the wild type was not statistically significant.

This was added in the text.

8. The authors have shown that addition of ATP causes an increase in calcium fluorescence intensity. This seems to be an unnecessary addition to the paper, since no further experiments were carried out to establish the potential role of ATP in linking calcium and ethylene mediated wound responses.

We disagree with the reviewer, as ATP signaling may be linked to ethylene signaling. One possible explanation for our observations is that ATP plays a role in facilitating ethylene pathway involvement in the calcium wave. While we did not further explore this connection in the current study, we believe that the ATP-induced increase in calcium fluorescence intensity provides relevant context for potential interactions between these pathways.

9. It has been shown that JA response to ablation is tissue specific. Provided that ethylene has an antagonistic effect JA signalling, it is important to show whether the ethylene responses also change in a tissue specific manner.

In our study, we observed that ethylene responses are robust and consistent across tissues following laser ablation.

Referee

#2

I have previously revised this manuscript when submitted to EMBO Journal, and read the point-by-point answers to my questions.

I am overall satisfied with the new version of the manuscript and how the authors have dealt with my requests.

I still have some doubts about the lack of a difference among the different backgrounds in terms of the maximum cytosolic Ca²⁺ peak (Fig. 1b). I can say that based on my experience, GCaMPs and R-GECO1 do have not a linear response. This means that over a certain threshold, in terms of Ca²⁺ concentration, the change in fluorescence is the same and relatively small differences, cannot be appreciated. This is one of the reasons why I suggested the authors use FRET-based ratiometric indicators. So, even if your data do not show a difference in the maximum Ca²⁺ peaks, it might be still present and could further support the authors' claims. Maybe, the authors can consider this aspect in the discussion.

Regarding the following answer: "...Regarding downstream responses, we currently lack an ideal method to directly monitor changes in calcium signatures and their impact on downstream signaling...".

A potential tool would be the use of CPKaleon sensors (Liese et al., Plant Cell 2023). A new CPKaleon sensor has been recently designed and created to support the effect of ABA-induced cytosolic Ca²⁺ in root cells (Lin et al., 2024 Nature Plants <https://doi.org/10.1038/s41477-024-01865-y>).

I am not asking you to develop a dedicated tool for your purposes, but this strategy could be discussed.

Thank you for suggestion. We included this point in the manuscript.

Aside from this initial comment, I have a few minor things.

Line 63. "...channel GLUTAMATE RECEPTOR-LIKE 3.3 (Bellandi et al, 2022)...". It would be correct to quote here the works Alfieri et al. 2020 (PNAS) and Grenzi et al., 2023 (Curr Biol) since they reported the properties of GLR3.3 ligand binding domain and its importance for the GLR3.3 Glutamate-dependent activation.

Thank you for suggestion. We made corrections accordingly.

Line 69. "...damage and it works in a similar pathway as GLRs...". Based on the model presented by Moe-Lange (Figure S15) it is more correct to say "...in a parallel pathway...".

Thank you for suggestion. We made corrections accordingly.

Line 194. "...induction was attenuated by $GdCl_3$, a Ca^{2+} channel blocker...". Please, keep in mind that Gd^{3+} is not a specific inhibitor of Ca^{2+} -permeable channels, being an inhibitor of non-selective cation channels.

Thank you for suggestion. We made corrections accordingly.

Line 220. "...ACC has been reported as the most effective ligand for inducing...". In Mou et al. authors do not show that ACC is bound by the GLR ligand binding domain so ACC is just a "predicted GLR ligand". This might indeed be the case (i.e. Grenzi et al., 2021 New Phytologist) but it has not been experimentally proved.

Thank you for suggestion. We made corrections accordingly.

Referee #3

The authors have addressed most of my previous concerns and have significantly extended the relevance of their findings by performing nematode bioassays in a series of channel mutants (*msl10*, *mca1/2*, *glr3.3/3.6*). The *msl10* and *glr* mutants (but not the *mca1/2* Ca^{2+} channel mutant) was more susceptible to nematode infections, providing the foundations for future molecular analyses. Overall, findings are timely and of broad interest to the plant science community.

At the same time, the revised version has not solved the following points:

1. Calcium is still placed both upstream and downstream of ET, with the authors concluding there is a dynamic feedback modulating the wound response. This claim is not fully supported by the data provided and may be a misleading conclusion due to how treatments were performed. Contrary to Ref.2 recommendations on toning down the calcium regulation and the lack of evidence in the literature that GLR3 and MSL10 are specific calcium channels, the authors decided to emphasize the role of calcium in ET and JA root production following laser ablation.

Our data suggest a dynamic feedback loop in the wound response, but we agree that this conclusion requires careful consideration, especially with regard to experimental design and the interpretation of the calcium-related treatments.

While we acknowledge the limitations in the specificity of GLR3 and MSL10 as calcium channels, as pointed out by Ref. 2, we argue that these channels play a significant role in calcium-mediated signaling during wound responses. However, we will tone down the language regarding their specificity and clarify that their roles are more likely to involve

modulation of calcium signaling rather than direct, exclusive involvement in calcium flux.

2. The negative effect of ET on JA responses has been explained in the rebuttal, but the new qPCR data is again showing only tendencies which are often not significant

We understand your concern regarding the qPCR data and its statistical significance. The qPCR analysis was performed using RNA extracted from a larger population of wounded roots in order to obtain enough material for reliable results. As a point of reference, Marhavy et al. (2019) demonstrated that the local ethylene response to laser ablation can reach concentrations around 300 μm , which we considered when interpreting our results.

We acknowledge that the qPCR data does not fully align with the results from the promoter reporter lines, and we agree that the tendency observed in our qPCR data lacks statistical significance. However, we have presented the data as it was obtained, while recognizing the limitations inherent in this type of analysis.

The complexity of hormone crosstalk, especially between ethylene and jasmonic acid, involves intricate regulation at multiple molecular levels, which makes it challenging to detect significant effects in every experimental condition. We appreciate the opportunity to clarify these aspects and will discuss the potential sources of variability and the complexity of hormone interactions in the revised manuscript.

3. Even if the authors address it in the rebuttal, the methods still do not report how were the Ca²⁺/ET/JA reporters introduced in different mutant backgrounds and which criteria were used to control for possible differences in transgene expression (also a concern from Ref.2). The authors should clearly indicate which generation (T1, T2, T3 or F1, F2, F3) is used under each figure and show results from more than one independent line for transformations.

We added this informations.

4. Most methods are written in protocol style and should be revised

We made corrections.

References

- Evans MJ, Choi W-G, Gilroy S & Morris RJ (2016) A ROS-Assisted Calcium Wave Dependent on the AtRBOHD NADPH Oxidase and TPC1 Cation Channel Propagates the Systemic Response to Salt Stress. *Plant Physiol* 171: 1771–1784
- Jewell JB & Tanaka K (2019) Transcriptomic perspective on extracellular ATP signaling: a few curious trifles. *Plant Signal Behav* 14: 1659079
- Lang T, Deng C, Yao J, Zhang H, Wang Y & Deng S (2020) A Salt-Signaling Network Involving Ethylene, Extracellular ATP, Hydrogen Peroxide, and Calcium Mediates K⁺/Na⁺ Homeostasis in Arabidopsis. *Int J Mol Sci* 21: 8683

- Marhavý P, Kurenda A, Siddique S, Dénervaud Tendon V, Zhou F, Holbein J, Hasan MS, Grundler FM, Farmer EE & Geldner N (2019) Single-cell damage elicits regional, nematode-restricting ethylene responses in roots. *EMBO J* 38: e100972
- Martin RE, Marzol E, Estevez JM & Muday GK (2022) Ethylene signaling increases reactive oxygen species accumulation to drive root hair initiation in *Arabidopsis*. *Development* 149: dev200487
- Tanaka K, Gilroy S, Jones AM & Stacey G (2010) Extracellular ATP signaling in plants. *Trends Cell Biol* 20: 601–608

Manuscript number: EMBOR-2024-60654V3

Title: Ca²⁺ Waves and Ethylene/JA Crosstalk Orchestrate Wound Responses in Arabidopsis Roots

Author(s): Xuemin Ma, Shamim Hasan, Muhammad Anjam, Saki Mahmud, Sabarna Bhattacharyya, Ute Vothknecht, Badou Mendy, Florian Grundler, and Peter Marhavy

Dear Peter,

Thank you for the submission of your revised manuscript to EMBO Reports. I have now evaluated your response to the referee concerns and all editorial changes and all seems fine. I am therefore writing with an 'accept in principle' decision, which means that I will be happy to accept your manuscript for publication once a few minor issues/corrections have been addressed, as follows.

As discussed, I noticed a few statements in the point-by-point response that require a reference and some clarification as to where this information can be found in the manuscript to further improve clarity of the document.

Once you have made these minor revisions, please use the following link to submit your corrected manuscript:

Link Not Available

If all remaining corrections have been attended to, you will then receive an official decision letter from the journal accepting your manuscript for publication in the next available issue of EMBO reports. This letter will also include details of the further steps you need to take for the prompt inclusion of your manuscript in our next available issue.

Thank you for your contribution to EMBO reports.

Kind regards,

Martina

All editorial and formatting issues were resolved by the authors.

Dr. Peter Marhavy
Swedish University of Agricultural Sciences
Department of Forest Genetics and Plant Physiology
Umeå Plant Science Centre
Umeå, Sweden 901 83
Sweden

Dear Peter,

I am very pleased to accept your manuscript for publication in the next available issue of EMBO reports. Thank you for your contribution to our journal.

Kind regards,

Martina
